 eLife

# Lymphatic endothelium stimulates melanoma metastasis and invasion via MMP14-dependent Notch3 and β1-integrin activation

Pirita Pekkonen[1†], Sanni Alve[1†], Giuseppe Balistreri[1], Silvia Gramolelli[1], Olga Tatti-Bugaeva[2], Ilkka Paatero[3], Otso Niiranen[1], Krista Tuohinto[1], Nina Perälä[1], Adewale Taiwo[1], Nadezhda Zinovkina[1], Pauliina Repo[2], Katherine Icay[2], Johanna Ivaska[3,4], Pipsa Saharinen[1,5], Sampsa Hautaniemi[2], Kaisa Lehti[2,6,7], Päivi M Ojala[1,7,8]*

[1]Research Programs Unit, Translational Cancer Biology, University of Helsinki, Helsinki, Finland; [2]Genome-Scale Biology, University of Helsinki, Helsinki, Finland; [3]Turku Centre for Biotechnology, University of Turku, Turku, Finland; [4]Department of Biochemistry, University of Turku, Turku, Finland; [5]Wihuri Research Institute, Helsinki, Finland; [6]Department of Microbiology, Tumor and Cell Biology, Karolinska Institutet, Stockholm, Sweden; [7]Foundation for the Finnish Cancer Institute, Helsinki, Finland; [8]Section of Virology, Division of Infectious Diseases, Department of Medicine, Imperial College London, London, United Kingdom

*For correspondence:
p.ojala@imperial.ac.uk

[†]These authors contributed equally to this work

Competing interests: The authors declare that no competing interests exist.

**Abstract** Lymphatic invasion and lymph node metastasis correlate with poor clinical outcome in melanoma. However, the mechanisms of lymphatic dissemination in distant metastasis remain incompletely understood. We show here that exposure of expansively growing human WM852 melanoma cells, but not singly invasive Bowes cells, to lymphatic endothelial cells (LEC) in 3D co-culture facilitates melanoma distant organ metastasis in mice. To dissect the underlying molecular mechanisms, we established LEC co-cultures with different melanoma cells originating from primary tumors or metastases. Notably, the expansively growing metastatic melanoma cells adopted an invasively sprouting phenotype in 3D matrix that was dependent on MMP14, Notch3 and β1-integrin. Unexpectedly, MMP14 was necessary for LEC-induced Notch3 induction and coincident β1-integrin activation. Moreover, MMP14 and Notch3 were required for LEC-mediated metastasis of zebrafish xenografts. This study uncovers a unique mechanism whereby LEC contact promotes melanoma metastasis by inducing a reversible switch from 3D growth to invasively sprouting cell phenotype.
DOI: https://doi.org/10.7554/eLife.32490.001

## Introduction

Distant organ metastasis requires that the tumor cells gain access into the hematogenous circulation (*Lambert et al., 2017*). The metastatic dissemination of cancer cells is expedited by molecular changes promoting the ability of cancer cells to invade across the surrounding extracellular matrices (ECM) and into the lumen of lymphatic or blood vessels. In order to survive in circulation as well as extravasate and colonize the distant organ sites the cancer cells need to survive a variety of stresses such as hemodynamic shear forces, trapping to vascular beds and ROS, and gain the ability to invasively grow at the new tissue microenvironment (*Piskounova et al., 2015*; *Strilic and Offermanns,*

**eLife digest** The death rates for many types of cancer have dropped, but melanoma remains a serious concern. This type of skin cancer is especially aggressive because it can spread to distant organs. Melanoma often spreads via the lymphatic system, a network of vessels that extends throughout the body to drain fluid from the body's tissues. The lymphatic system also includes structures – called lymph nodes – that filter bacteria from this fluid; this helps to defend against infection.

When melanoma spreads to lymph nodes and distant organs, clinicians diagnose it as Stage IV melanoma. For patients at this stage, the outcome is often poor. It is clear that melanoma exploits lymph vessels to spread throughout the body. But researchers also suspect that vessel cells interact with the cancer cells, helping the melanoma invade distant organs. Understanding exactly how lymph vessels promote the spread of melanoma will lead to better options for treating this aggressive cancer.

Pekkonen, Alve et al. investigated whether exposing human melanoma cells to cells from the walls of human lymph vessels would make the cancer cells more aggressive. Indeed, after growing the two cell types together in the laboratory, the melanoma cells became more invasive. When transplanted into mice, these cancer cells spread to and invaded the rodents' distant organs.

Pekkonen, Alve et al. conducted a series of experiments to identify specific proteins in the melanoma cellsthat were responsible for making the cancer more invasive after it interacted with the lymph vessel cells. These experiments identified proteins called MMP14, Notch3, and β1-integrin as critical to the invasive spread of melanoma cells. When melanoma cells with less MMP14 or Notch3 were implanted into zebrafish, the cancer cells spread less efficiently. These findings may represent new leads that clinicians can test to see if they are markers of cancers that are most likely to spread and that the pharmaceutical industry can pursue to treat melanoma patients.

DOI: https://doi.org/10.7554/eLife.32490.002

*2017*). Many of these activities have been found to be enhanced in tumor cell clusters relative to singly invading tumor cells.

The local microenvironment including the ECM and cell-non-autonomous interactions between cancer cells and stromal cells play a vital role in metastasis. In addition to providing a direct route for dissemination, the tumor lymphatics have been proposed to directly modulate the metastatic cascade through mechanisms that have remained elusive (*Alitalo and Detmar, 2012*). Clinical observations of satellite melanoma tumors growing between the primary tumor and draining lymph nodes have suggested that the surrounding lymphatic endothelium serves as a protective microenvironment for the survival of incipient metastatic cells (*Meier et al., 2002*). In support of this hypothesis, over the last few years it has become increasingly clear that lymphatic endothelial cells (LECs) in fact actively interact with the surrounding cells in the tissue, thus regulating both physiological and pathological processes including tumor progression and metastasis. Both paracrine communication and direct cell-cell interactions between tumor cells and the associated lymphatics have been shown to drive tumor progression and dissemination. For example, chemokine receptor-ligand interactions between melanoma and LECs drive chemotaxis of tumor cells towards the lymphatics (*Cabioglu et al., 2005*; *Das et al., 2013*; *Shields et al., 2007*). Alternatively, tumor cells secrete factors like lipoxygenase, which can induce downregulation of the endothelial surface molecules and loosening of the LEC junctions in vitro (*Kerjaschki et al., 2011*).

In this study, we set to investigate how the LECs in the tumor microenvironment affect the metastatic melanoma cell phenotype. To this end, we implemented 2D- and 3D melanoma-LEC co-culture models, which enable a systematic analysis of the molecular crosstalk between the tumor cells and the lymphatic endothelium. We found that the interaction of melanoma cells with LECs induced matrix-metalloproteinase-14 (MMP14, also known as MT1-MMP) -dependent Notch3 and β1-integrin activation in the expansively growing metastatic melanoma cells, leading to invasive sprouting of cells in 3D matrices. Importantly, the interaction of these melanoma cells with LECs led to significantly increased metastasis of melanoma xenografts in vivo, which was dependent on MMP14 and

Notch3. Thus, the crosstalk with LECs promotes melanoma metastasis by inducing a reversible switch to invasively sprouting melanoma cells.

## Results

### Three-dimensional co-culture model recapitulates the melanoma interaction with lymphatic vasculature

To study the interaction of LECs with melanoma cells in vitro, we utilized a three-dimensional (3D) co-culture method, where LEC spheroids were embedded together with single, GFP-expressing or fluorescent tracer labeled, melanoma cells into a cross-linked 3D matrix (*Figure 1a*) (*Korff and Augustin, 1998*; *Tatti et al., 2015*). We chose to use fibrin as 3D matrix since it is frequently deposited within tumor tissues and perivascular spaces in vivo. After 72 hr incubation, the LEC control spheroids (labeled with the endothelial marker PECAM) showed moderate outgrowth (LEC sprouting) from the spheroid body (*Figure 1b*, leftmost panel). Co-culturing the LEC spheroids with cells isolated from a melanoma skin metastasis (WM852) or from a superficially spreading melanoma (Bowes), resulted in melanoma attraction and invasion into the LEC spheroid (*Figure 1b*, middle and rightmost panels). Especially the WM852 cells (and Bowes to a lesser extent) appeared to disrupt the spheroid structure at the melanoma cell-LEC contact sites, as shown by the loss of the endothelial adhesion molecule PECAM (*Figure 1b*, enlarged inserts of the middle and rightmost panels). Thus, our 3D co-culture model qualitatively recapitulates attraction, migration and invasion of melanoma cells into lymphatic endothelial structures.

### Interaction with LECs increases the metastasis of melanoma cells in vivo

To investigate potential crosstalk between melanoma cells and LECs and effects of these interactions on melanoma tumorigenesis in vivo, we isolated cells from 3D co-cultures. For this, the GFP and luciferase expressing WM852 and Bowes cells were cultured in 3D as monotypic, single cell suspension or together with the preformed LEC spheroids for 72 hr after which protease inhibitors were removed for 30–48 hr leading to 3D matrix digestion and release of the cells. To quantify LECs in the recovered cell mixtures, the cells were subjected to a qRT-PCR analysis for the expression of the LEC markers *CD34*, *PROX1* and *FLT4* (gene for VEGFR3). Parental primary LECs were used as a control. The cells derived from the 3D co-cultures were essentially negative for these LEC markers (*Figure 1—figure supplement 1a*), indicating that the cell isolation procedure favored the enrichment and survival of the melanoma cells. We therefore named these initially co-cultured melanomas as LEC primed WM852* or Bowes* (distinguished by asterisks from the parental cells derived from monotypic cultures).

Next, LEC primed WM852* or Bowes*, or WM852 or Bowes from monotypic cultures as controls, were subcutaneously implanted into SCID mice (*Figure 1a*). LEC priming did not significantly affect the growth rate of the WM852 primary tumors (*Figure 1c*). Similarly, the growth rate of the 3D LEC primed Bowes tumors was equal to the Bowes tumors derived from the monotypic cultures (*Figure 1d*), although the tumor volume and weight were slightly higher in the 3D LEC primed Bowes tumors over the monotypic Bowes tumors at the end point analysis (*Figure 1—figure supplement 1b*).

Subsequent analyses of the WM852* or Bowes* derived tumors revealed melanoma cell invasion into the lymphatic vessels in a manner similar to the in vitro 3D co-cultures (*Figure 1—figure supplement 1c*). To assess whether the LEC priming of melanoma cells affected their metastatic capacity in vivo, we imaged lymph nodes, lungs and livers isolated from the mice bearing WM852/WM852* or Bowes/Bowes* derived tumors.

Mice implanted with monotypic WM852 cells, originating from a melanoma metastasis, showed clearly stronger luciferase signal in the lymph nodes than the Bowes groups (*Figure 1—figure supplement 1d–e*) but only low levels of signal in liver and lungs (*Figure 1e–f*). In contrast, the LEC primed WM852* tumors metastasized significantly to both liver and lungs (*Figure 1e–f*). Supporting the increased distant organ metastasis, quantitative PCR from the mouse lung genomic DNA revealed higher amounts of the human-specific Alu sequences in mice bearing the WM852* tumors when compared to the lungs derived from the monotypic WM852 implanted mice (*Figure 1—figure supplement 1f*).

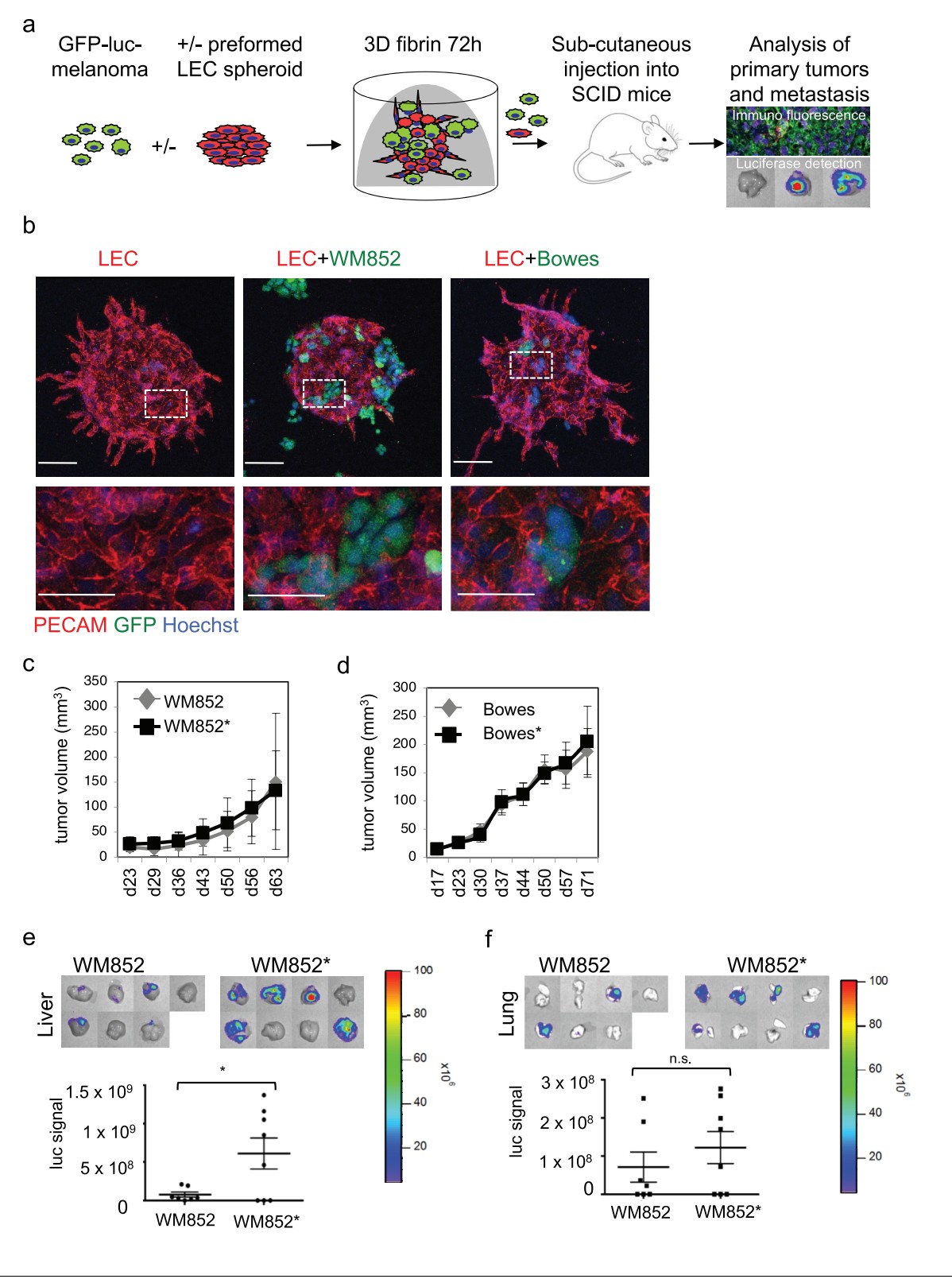

**Figure 1.** Co-culture of melanoma cells and LECs reveals melanoma invasion into the LEC 3D structures and increases the metastatic potential of WM852 cells in vivo. (a) Schematic of the experimental pipeline. (b) Confocal images of LEC spheroids (PECAM-1, red) in 3D fibrin matrix (left panel), LEC spheroids co-cultured with WM852 (green, middle panel) or Bowes (green, left panel). The area enclosed in the white square is shown enlarged below each panel. Melanoma cells were stained with GFP (green), and nuclei were counterstained with Hoechst 33342. Maximum intensity Z-projections

*Figure 1 continued*

of confocal stacks are shown. (**c,d**) Growth rates of the 3D LEC primed WM852* (**c**) and Bowes* (**d**) derived tumors (n = 8 for both cell types) compared to control WM852 (n = 7) and Bowes (n = 8) tumors, respectively. (**e, f**) Distant organ metastasis, detected by bioluminescence imaging of luciferase signal, in liver (**e**) and lung (**f**) of SCID mice subcutaneously injected with WM852 alone or co-cultured with LECs (WM852*). Upper panels: representative images of the indicated organs, each box represents an organ from one mouse. Bottom panel: quantification of luciferase signal, each dot represents the luciferase value in one sample. Horizontal line indicates the average, vertical bars represent SEM. *: p<0.05. n.s., non-significant.

DOI: https://doi.org/10.7554/eLife.32490.003

The following figure supplement is available for figure 1:

**Figure supplement 1.** Analysis of mouse xenografts and distant organ metastasis.

DOI: https://doi.org/10.7554/eLife.32490.004

In concordance with the non-metastatic origin of the Bowes cells, mice with monotypic Bowes or Bowes* had luciferase positive tumor cells in few of the isolated lymph nodes (*Figure 1—figure supplement 1e*) and no significant metastasis to liver or lungs (*Figure 1—figure supplement 1g*).

These results indicate that the in vitro interaction of WM852 metastatic melanoma cells with LECs prior to tumor implantation promotes distant organ metastasis in vivo.

## Interaction with LECs induces transcriptional changes in melanoma gene expression

To enable functional and molecular analysis of the changes occurring in melanoma cells and LECs upon the co-culture, we utilized a 2D co-culture model and optimized a separation method for the two cell types. The GFP-melanoma cells were loaded with dextran-coated magnetic nanoparticles prior to the 2D co-culture with LECs. After co-culture for 24–48 hr, LECs and the primed melanoma cells were isolated using magnetic columns and the separation was validated with antibodies and qRT-PCR (workflow depicted in *Figure 2—figure supplement 1a*; validations *Figure 2—figure supplement 1b–c*), confirming efficient separation of the two cell populations: isolated WM852* showed only 0.1–1% of LEC marker expression (*Figure 2—figure supplement 1c*, left panel). The separation of Bowes* was slightly less efficient (*Figure 2—figure supplement 1c*, right panel). No differences were observed in the proliferation of LEC-primed, separated WM852* and Bowes cells* when compared to cells from the corresponding monotypic cultures (*Figure 2—figure supplement 1d*).

We next subjected the ±LEC primed, separated WM852 and Bowes cells to RNA sequencing (*Figure 2—figure supplement 1a*). When compared to the parental cells derived from monotypic cultures, expression of 663 genes was upregulated (>2 fold change, p<0.05) and expression of 347 was downregulated (<2 fold change, p<0.05) in the LEC primed WM852* cells (*Supplementary file 1*, related to *Figure 2*). The LEC primed Bowes* cells showed 532 upregulated genes and 14 downregulated genes (2-fold change, p<0.05) when compared to Bowes from monotypic cultures (*Supplementary file 1*, related to *Figure 2*). Thus, LEC interaction resulted in differential expression of a large number of genes in the LEC primed melanoma cells.

To further dissect the biological processes affected by LEC interaction, we next utilized Generally Applicable Gene-set Enrichment (GAGE) for pathway analysis. Interestingly, LEC priming led to enrichment of several pathways known to be involved in cancer metastasis as well as cell contacts and communication (*Figure 2a–b*). Several of these pathways were enriched in both the metastatic cell line WM852 as well as non-metastatic Bowes cells after the LEC contact. The commonly upregulated pathways after LEC priming included adherens junctions, regulation of actin cytoskeleton, Notch signaling and gap junctions (*Figure 2a–b*; red text), whereas ECM-receptor signaling was downregulated in WM852* cells and upregulated in Bowes* cells (*Figure 2a–b*, blue text). In addition, other pathways involved in cell-cell and cell-matrix interactions, like focal adhesion, TGF-β signalling and tight junction pathways, were enriched as differentially regulated only in Bowes cells after the LEC contact (*Figure 2a–b*, black text).

To identify genes enriched in the pathways involved in the cell-cell or cell-matrix contacts (focal adhesion, regulation of actin cytoskeleton, adherens junction, gap junction, tight junction, ECM-receptor interaction and TGF-β signaling), we selected significant, differentially expressed genes in WM852* and Bowes* cells and sorted them to the pathways (*Supplementary file 2*, related to *Figure 2*). We found LEC-induced changes in melanoma cells in the expression of ECM matrix components such as collagens, laminins, fibronectin and reelin; cell surface receptors such as integrins,

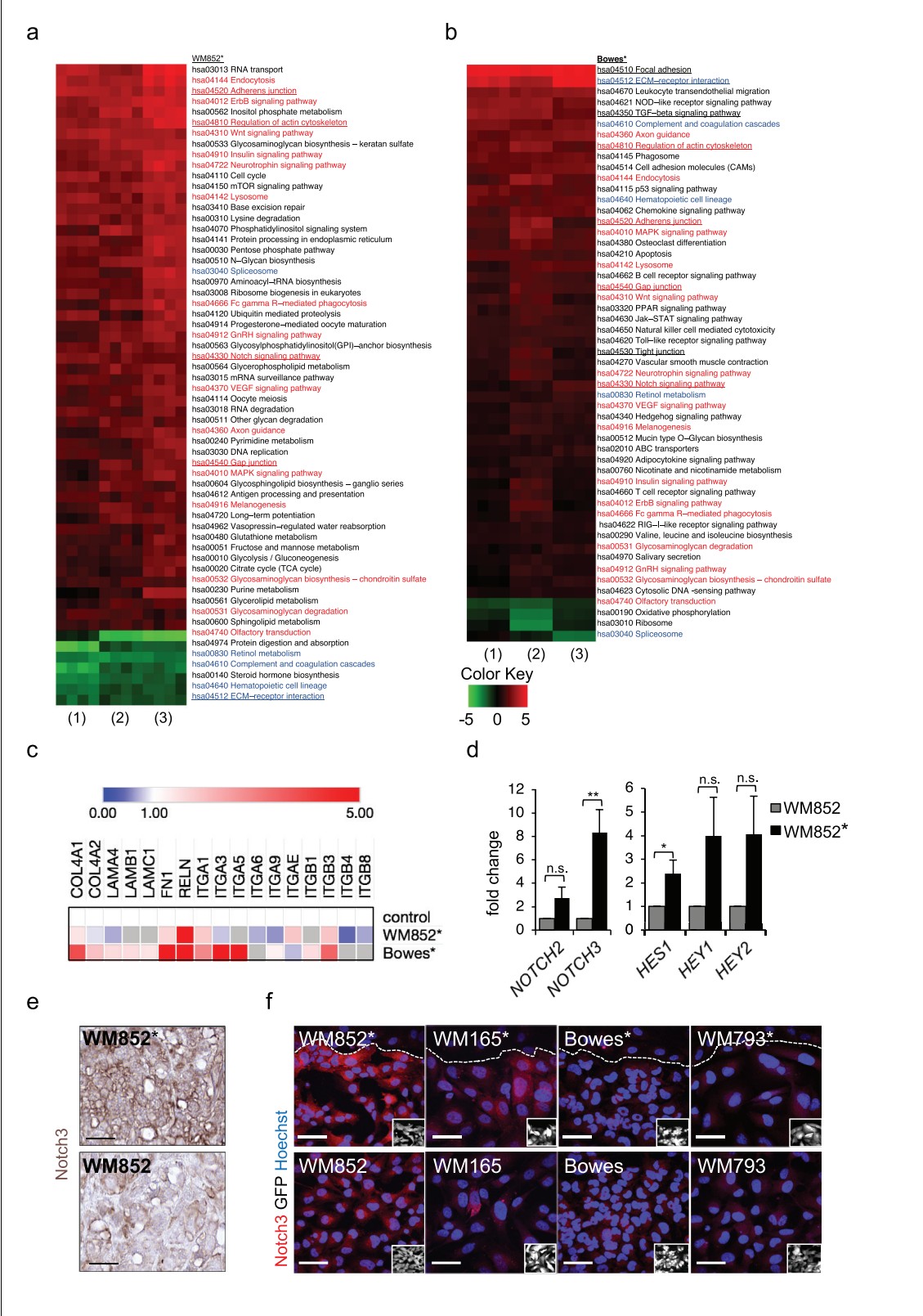

**Figure 2.** Transcriptomic analysis reveals Notch3 upregulation in the LEC primed WM852 cells. (**a–b**) Generally Applicable Gene-set Enrichment (GAGE) for RNA-seq pathway analysis of LEC primed (**a**) WM852* and (**b**) Bowes* cells. Samples were compared to their respective cells derived from monotypic cultures. Three biological replicates per sample group and four run replicates were used. In the heatmap, red represents upregulated and green downregulated pathways in WM852* and Bowes*. Pathways enriched in both cell lines are marked with red text if they were similarly upregulated

*Figure 2 continued on next page*

*Figure 2 continued*

or downregulated, and blue text if they were differentially upregulated or downregulated. Pathways enriched only in one cell line are marked with black text. Pathways with underlined text were used for further analysis. (c) Heatmap depicting average expression fold change of the differentially expressed ECM-receptor interaction pathway genes in the RNA-seq of LEC primed WM852* and Bowes* cells. The WM852 and Bowes cells from monotypic cultures were used as controls, and set to one. Red represents upregulated and blue downregulated genes in WM852* and Bowes*. Adjusted p-values are less than 0.05 for all genes shown. (d) Relative mRNA fold change of the indicated targets in WM852 and WM852*. *:p<0.05; **p<0.001; n.s., non-significant. (e) Representative images of Notch3 immunohistochemistry in the WM852 and WM852* derived xenografts. Scale bar = 50 μm. (f) Representative confocal images of Notch3 staining (red) in different GFP expressing melanoma cell lines (GFP positive cells shown in the inset) cultured in the presence (*, upper panels) or absence (bottom panels) of LECs. Nuclei are counterstained with Hoechst 33342. The dashed line indicates the LEC-melanoma (below the line) border. Scale bar = 50 μm. Full size confocal images are available as a *source data 1*.

DOI: https://doi.org/10.7554/eLife.32490.005

The following figure supplements are available for figure 2:

**Figure supplement 1.** Optimization of magnetic separation of melanoma and LEC cells following co-culture of LEC and melanoma cells.

DOI: https://doi.org/10.7554/eLife.32490.006

**Figure supplement 2.** LEC priming induced differential changes in the ECM-receptor interaction pathway and Notch signaling pathway.

DOI: https://doi.org/10.7554/eLife.32490.007

TGF-beta receptors and platelet derived growth factor receptors; as well as actin cytoskeleton (*Figure 2c*, *Supplementary file 2* related to *Figure 2*). Notably, the ECM-receptor interaction pathway was the most differentially regulated pathway in the two cell lines (*Figure 2—figure supplement 2a–b*). Majority of the genes in this pathway were upregulated in Bowes* cells when compared to parental Bowes cells, whereas in WM852* cells these were mainly downregulated (*Figure 2c*, see examples of differentially expressed genes, *Supplementary file 2* related to *Figure 2*). Taken together, LEC priming induces major changes in the genes of the cell-cell and cell-ECM contact mediator pathways that partly differ between the metastatic WM852* and the non-metastatic Bowes* cells.

## LEC interaction induces Notch3 in WM852 cells

Our pathway enrichment analysis revealed Notch signaling pathway to be upregulated in the melanoma cells after LEC priming (*Figure 2a–b*). Previous studies using co-culture of HUVECs and melanoma cells have identified Notch3 as a player in melanoma–EC communication (*Stine et al., 2011*) and a potential mediator of melanoma metastasis also in human tumors (*Howard et al., 2013*). We therefore next investigated if Notch3 and its downstream targets were induced in the LEC primed melanomas. In the RNA-sequencing analysis, the upregulated genes in the LEC primed WM852* cells included several Notch pathway members including *NOTCH3* and the Notch downstream target *HEY1* (*Figure 2—figure supplement 2c*), and qRT-PCR analyses confirmed elevated *NOTCH3* levels accompanied by increases in the known Notch downstream targets (*HES1*, *HEY1,* and *HEY2*) in WM852* when compared to the monotypic WM852 (*Figure 2d*). In Bowes* cells a two-fold upregulation of *HES1*, was detected (*Figure 2—figure supplement 2d*).

Importantly, upregulation of Notch3 was also evident in WM852* tumors by immunohistochemistry when compared to tumors derived from the monotypic WM852 (*Figure 2e*, *Figure 2—figure supplement 2e*). The upregulation of Notch3 upon LEC priming provides strong support that our LEC-melanoma interaction model can reveal clinically relevant molecules for cancer dissemination and thus has high potential also for identifying previously unrecognized pathways and molecules contributing to the lymphatic metastasis of melanoma.

To address if the LEC induced *NOTCH3* increase was specific for WM852 melanoma cells, we co-cultured two additional melanoma cell lines derived from vertical growth phase primary tumor (WM793) and metastasis (WM165) (*Tatti et al., 2011*) with LECs and analysed the co-cultures for Notch3 expression by indirect immunofluorescence analysis (IFA). LEC priming induced an increase in Notch3 expression specifically in the metastatic cell lines, namely WM852 and WM165, but not in the primary tumor derived WM793 or Bowes (*Figure 2f*; *Figure 2—figure supplement 2f*). Additional qRT-PCR analysis revealed the most pronounced upregulation of *NOTCH3* in the primed metastatic cell lines (*Figure 2—figure supplement 2g*).

## Interaction with LECs switches on invasive melanoma growth in 3D matrix

To gain more insight into the metastasis-promoting mechanisms of LECs on melanoma cells, we investigated their ability to grow in 3D fibrin matrix. Matrix embedded WM852* and WM165* cells displayed a different morphology forming string-like, sprouting cell colonies (*Figure 3a*), and resembling cells undergoing collective invasion. In contrast, the non-primed control cells from monotypic cultures formed round sphere-like colonies in fibrin (*Figure 3a*). The morphology of the non-metastatic cell lines instead remained unaffected by priming. Bowes cells grew as sparse, elongated cells and WM793 as collective sphere-like cell clusters in fibrin (*Figure 3a,b*), suggesting that both expansive growth and invasive sprouting are important for aggressive dissemination.

To assess the duration of the LEC-induced change in the 3D growth phenotype of melanoma cells, the WM852* cells were cultured in 2D monotypic cultures for 0, 1, 3 and 7 days before embedding in 3D fibrin for 96 hr. WM852 cells derived from a monotypic culture at day 0 were used as a control. The sprouting phenotype of WM852* cells was retained up to 3 days after separation and declined back to control levels after 7 days (*Figure 3c*). The ability of LECs to prime melanomas and induce their sprouting required direct contact between the two cell types and was not mediated by paracrine factors secreted by the LECs since conditioned medium (CM) from LEC monoculture or WM852-LEC co-culture failed to induce sprouting of matrix embedded WM852 cells (*Figure 3d*). These results indicate that the LEC-induced changes in WM852 are transient and require a prior direct cell-cell contact with the LECs.

## LEC-induced metastatic melanoma 3D growth phenotype is Notch3 dependent

Next we assessed the requirement of Notch3 for the change in the 3D growth phenotype of the LEC primed WM852. To this end, we inhibited Notch activation by treating the WM852 monotypic culture or the LEC-WM852 co-culture with DAPT, an inhibitor of gamma secretase that mediates the cleavage of Notch receptors to produce the active intracellular domain form (NICD). After magnetic separation, the cells were subjected to the 3D fibrin assay in the presence of DAPT or vehicle (ethanol; EtOH). The DAPT treatment led to a dramatic reduction in the relative sprouting index of the LEC primed WM852 cells but had little effect on the control WM852 derived from monotypic cultures (*Figure 3—figure supplement 1a*). To demonstrate that the DAPT effect was specific for Notch3 inhibition, we repeated the assay by treating the cells with siRNA targeting *NOTCH3* (*Figure 3—figure supplement 1b*). Notch3 depletion almost completely abolished the increase in the sprouting growth of the LEC primed WM852 (*Figure 3e*), further supporting the role of Notch3 for the switch to the sprouting 3D growth of melanoma cells induced by LEC interaction.

## MMP14 is required for the invasively sprouting 3D growth of LEC primed melanoma cells

Membrane type matrix metalloproteinase MMP14 is frequently induced in invasive melanoma and its high expression correlates with melanoma progression and metastasis (*Hofmann et al., 2000*; *Tatti et al., 2015*). Therefore, we investigated whether LEC priming would induce changes in MMP14 expression in melanoma cells. Interestingly, higher MMP14 signal intensity and re-localization to the cell-cell contacts/plasma membrane were observed in WM852* (*Figure 4a*, arrowheads). The increase in the cell surface MMP14 protein levels upon LEC co-culture was also confirmed by flow cytometry analysis for WM852 (*Figure 4—figure supplement 1a*). A moderate increase in the MMP14 intensity was observed in WM165* (*Figure 4—figure supplement 1b*), but no LEC-induced changes in the level or localization of MMP14 were observed in Bowes (*Figure 4a*) or WM793 cells (*Figure 4—figure supplement 1b*). Bowes cells in particular showed a strong perinuclear MMP14 signal, which co-localized with a signal from anti-TGN46 that defines the location of trans-Golgi network (*Figure 4—figure supplement 1c*).

The potential role of MMP14 for the invasively sprouting 3D growth phenotype of WM852* was analysed by subjecting 3D fibrin-embedded cells to a pan-MMP inhibitor GM6001 (*Figure 4—figure supplement 1d*) or *MMP14* silencing with two different targeting siRNAs or a non-targeting control siRNA (siCtrl) (*Figure 4—figure supplement 1e*). Quantification of the relative sprouting index demonstrated that both GM6001 (*Figure 4—figure supplement 1d*) and siMMP14 treatments

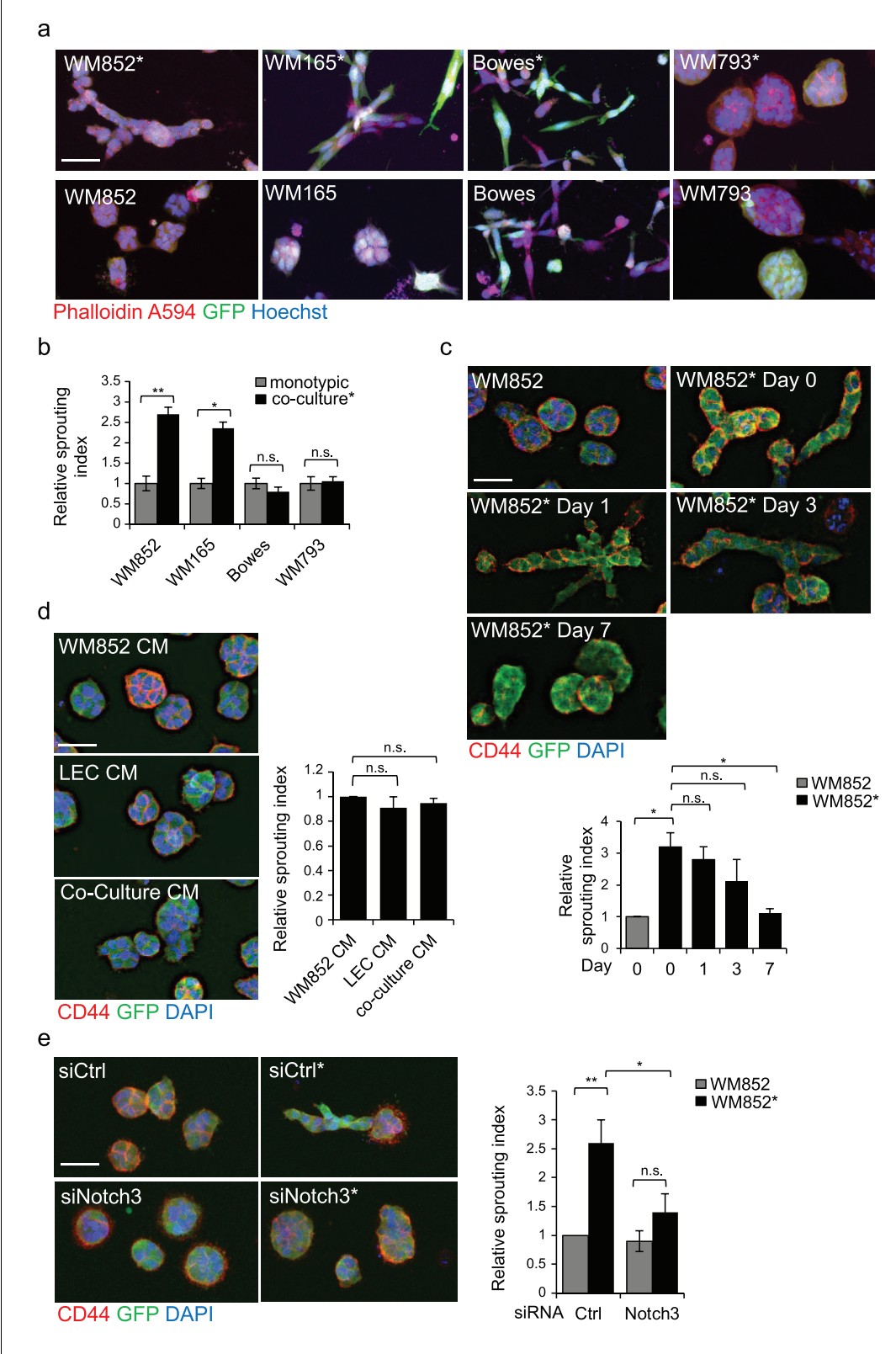

**Figure 3.** LEC interaction modifies the 3D growth phenotype of the melanoma cells. (a) Representative confocal images of 3D fibrin assays after magnetic separation of the indicated melanoma cell lines co-cultured with LEC (*, upper panels) or from monotypic culture (bottom panels). GFP expressing (green) melanoma cells were stained with Phalloidin A594 (red), nuclei are counterstained with Hoechst 33342 (blue). Maximum intensity Z-projections of the confocal stacks are shown. Scale bar = 50 μm. (b) Quantification of the sprouting index for the samples in (a). Graphs show the

*Figure 3 continued on next page*

*Figure 3 continued*

average of at least three images per condition per two independent experiments, error bars indicate the SEM p<0.05; **; p<0.01; n.s., non-significant. (c) Representative images of the 3D fibrin assay of WM852 and WM852* after magnetic separation at the indicated day after separation. The graph represents the average of three images per condition analysed in each of the two independent experiments. Error bars represent SEM. *: p<0.05; n.s., non-significant. (d) Representative images of the 3D fibrin assay of monotypic WM852 treated with conditioned media (CM) from the indicated sources. The graph represents the average of three images per condition. Error bars represent SEM. n.s., non-significant. (e) Representative images of the 3D fibrin assay of WM852 and WM852* treated with the indicated siRNAs for 72 hr prior to magnetic separation and fibrin embedding. The graph represents the average of three images per condition per three independent experiments. Error bars represent SEM. *: p<0.05; **; p<0.01; n.s., non-significant.

DOI: https://doi.org/10.7554/eLife.32490.008

The following figure supplement is available for figure 3:

**Figure supplement 1.** 3D growth phenotype of metastatic melanoma cells is Notch3 dependent.

DOI: https://doi.org/10.7554/eLife.32490.009

(*Figure 4b*) almost completely abolished the sprouting of LEC primed WM852 in 3D, indicating that the LEC-induced change in the growth phenotype was dependent on MMP14.

Given that the 3D sprouting of WM852* cells was also inhibited by *NOTCH3* silencing, we first assessed MMP14 and Notch3 co-localization in WM852* by IFA. In WM852* (*Figure 4c*, arrowheads, left panel, and *Figure 4—figure supplement 2a* upper panels for a close up including the channels for MMP14 and Notch3 stainings), the two proteins were expressed on the plasma membrane with occasional overlap in the cell-cell contacts, while in WM852 no co-distribution was observed (*Figure 4c*, right panel and *Figure 4—figure supplement 2a*, bottom panels). Since we and others have previously shown that MMP14 transcription and protein expression are induced by Notch signaling in other cell types (*Cheng et al., 2011*; *Funahashi et al., 2011*), we next assessed if MMP14 and Notch3 were co-regulated in WM852*. Depletion of *MMP14* mRNA by 98% significantly reduced the mRNA level of *HEY1* and resulted in a non-significant decrease of *NOTCH3*, but, however, had no effect on *HES1* mRNA (*Figure 4d*), and reduced Notch3 signal in the WM852* cells (*Figure 4e–f*), indicative of co-regulation. However, efficient depletion of *NOTCH3* (by 94%) or *HEY1* (by 80%) had no effect on *MMP14* mRNA (*Figure 4—figure supplement 2b*) or MMP14 protein levels (*Figure 4—figure supplement 2c*). To study the co-regulation further, we treated the co-cultures of LEC and WM852 with a MMP14 specific inhibitor (NSC 405020), after which the WM852 cells were separated for further analysis. NSC 405020 treatment reduced expression of the full length and active cleaved Notch3 (NICD3) by 42% in the WM852* cells (*Figure 4—figure supplement 2d–e*). These data further support that MMP14 positively regulates Notch3 expression and activation, thereby contributing to the change in the 3D growth phenotype of the LEC primed WM852.

## Change in the 3D growth of LEC primed melanoma cells is integrin dependent

Our trancriptomic analysis revealed changes in pathways involved in cancer invasion and metastasis including several cell adhesion pathways (*Figure 2a–b* and *Supplemental file 2*). We therefore decided to address the role of integrins, one of the major cell-matrix adhesion molecule families, in the LEC induced changes in metastatic melanoma cells. Since several reports support association of β1-integrin expression with melanoma progression and metastasis (*Danen et al., 1994*; *Kato et al., 2012*; *Natali et al., 1993*), we investigated the expression and activation state of β1-integrin in the ±LEC co-cultured WM852, WM165, WM793, and Bowes cell lines using antibodies for the total and active β1-integrin (12G10). Interestingly, WM852* showed a higher signal for the active β1-integrin (*Figure 5a* and *Figure 5—figure supplement 1a*), which was further confirmed by staining with another antibody against active β1-integrin, 9EG7 (*Figure 5—figure supplement 1b*). We also attempted to confirm the integrin activation in WM852* by FACS analysis using antibodies against active β1-integrin (12G10 and 9EG7). However, we did not detect the increase in active β1-integrin levels in the FACS of WM852* (data not shown) perhaps because the activation may either require the constant contact of melanoma cells to LECs or to 3D fibrin, and be sensitive to the cell detachment process. Also WM165* cells displayed an increase in the active β1-integrin signal intensity, which however did not reach statistical significance (*Figure 5a* and *Figure 5—figure supplement*

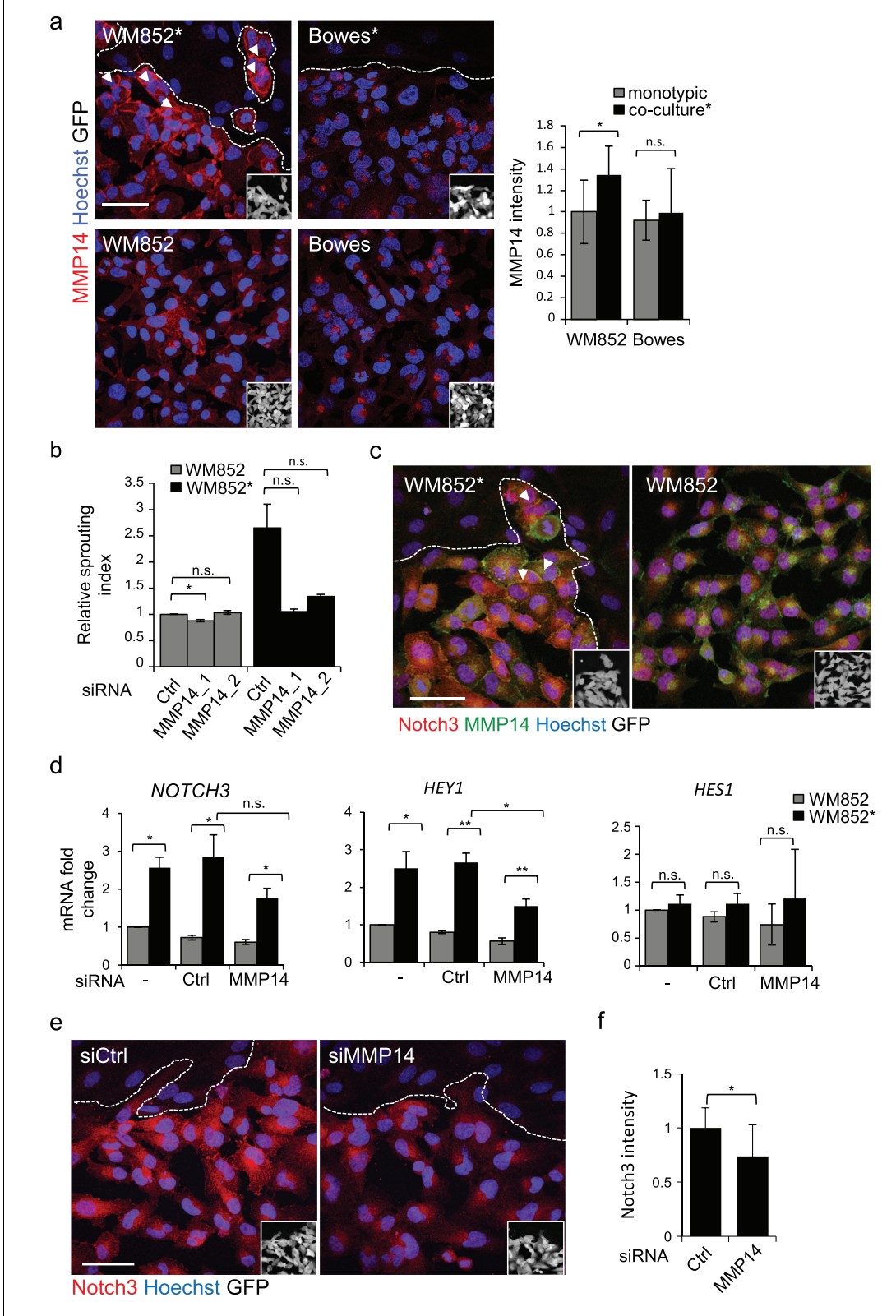

**Figure 4.** MMP14 is required for the increased sprouting growth of LEC primed melanoma cells in 3D. (a) Right panels: representative confocal images of MMP14 (red) expression in WM852 and Bowes co-cultured with LECs (*, upper panels) and from monotypic culture (bottom panels). Nuclei were counterstained with Hoechst 33342. GFP-expressing melanoma cells are shown white in the inset. Arrowheads indicate MMP14 localization to the cell-cell contacts. The dashed line indicates the LEC-melanoma (below the line) border. Scale bar = 50 μm. Left panel: quantification of MMP14 intensity

*Figure 4 continued on next page*

*Figure 4 continued*

analysed in four images per condition from two independent experiments. More than 100 cells were always analysed per condition. Average is shown, error bars represent SD; *: p<0.05. n.s., non-significant. (**b**) Quantification of the 3D sprouting index of WM852 and WM852* treated with the indicated siRNAs for 72 hr followed by magnetic separation and the 96 hr fibrin assay. The graph represents the average of three images per condition in each of the two independent experiments, error bars indicate SEM; *: p<0.05. n.s., non-significant. (**c**) Representative confocal images of MMP14 (green) and Notch3 (red) in WM852 and WM852*. Arrowheads indicate the cell-cell junction where MMP14 and Notch3 co-localize. Nuclei were counterstained with Hoechst 33342. The dashed line indicates the LEC–melanoma (below the line; GFP positive cells (white) in the inset) border. Scale bar = 50 µm. (**d**) mRNA fold change of the indicated targets in WM852 and WM852* upon treatment with the indicated siRNA for 72 hr and following magnetic separation. Graphs show the average of three independent experiments, error bars indicate SEM, *: p<0.05; **: p<0.01. n.s., non-significant (**e**) Representative confocal images of Notch3 staining (red) in WM852* treated with the indicated siRNAs for 72 hr. Nuclei were counterstained with Hoechst 33342. The dashed lines indicate the LEC-WM852 (GFP positive cells (white) in the inset) border. Scale bar = 50 µm. (**f**) Quantification of Notch3 signal intensity of WM852* treated as in (**e**) and described in (**a**). Error bars indicate SD; *: p<0.05. Full size confocal images are available as *source data 2*.

DOI: https://doi.org/10.7554/eLife.32490.010

The following figure supplements are available for figure 4:

**Figure supplement 1.** MMP14 levels increase and MMP14 activity is needed for 3D sprouting growth in LEC-primed metastatic melanoma.

DOI: https://doi.org/10.7554/eLife.32490.011

**Figure supplement 2.** Notch3 does not regulate MMP14 expression, but MMP14 positively regulates Notch3.

DOI: https://doi.org/10.7554/eLife.32490.012

---

*1a*). In contrast, no changes in the active β1-integrin levels were observed with the WM793 and Bowes cell lines ± LECs (*Figure 5a* and *Figure 5—figure supplement 1a*). When we analysed total β1-integrin by IFA, no detectable differences in distribution or signal intensity in any of the cell lines ± LEC by IFA were found (*Figure 5—figure supplement 1c–d*). Intriguingly, total β1-integrin, detected by immunoblotting, was decreased markedly in the LEC co-cultured WM852* cells (by 43%), and about 25% (although non-significant) for WM165* and Bowes (*Figure 5—figure supplement 1e*). In addition, the LEC co-culture altered the proportion of mature β1-integrin in WM852* and WM165* cells since the upper β1-integrin specific band (mature form of the integrin) decreased and a faster migrating, probably representing newly synthesized, immature β1-integrin appeared. Notably, the electrophoretic pattern of β1-integrin was similar in WM852 and WM793 cells before and after the interaction with LEC especially when compared to the pattern in Bowes cells, possibly reflecting differences in the processing and trafficking of β1-integrin. The discrepancy between the IFA and immunoblotting data may be due to different accessability of antibody-epitopes, and the potential shift in integrin processing suggested by the immunblot warrants further investigation in future studies.

To test if the LEC-induced integrin activation was contributing to the invasively sprouting 3D growth of WM852* cells, separated cells were subjected to the 96 hr fibrin assay in the presence of a β1-integrin blocking antibody AIIB2. The treatment almost completely abolished the LEC induced change in the growth phenotype of WM852 cells as compared to untreated cells (*Figure 5b*). This result indicates that also β1-integrin contributes the sprouting growth of WM852* cells.

## MMP14 is the upstream regulator of Notch3 and β1-integrin

MMP14 localization to β1-integrin containing adhesion complexes has been demonstrated during cancer cell invasion process (*Woskowicz et al., 2013*), (*Vuoriluoto et al., 2011*). We therefore first addressed the β1-integrin and MMP14 localization in WM852* cells. Both MMP14 and the active β1-integrin localized onto plasma membrane (*Figure 5c*). Given that MMP14 directly associates with β1-integrin and controls its expression levels in the branching morphogenesis of mammary epithelium (*Mori et al., 2013*), we decided to assess the role of MMP14 in the integrin activation. *MMP14* silencing in WM852 cells, prior to their co-culture with LECs, significantly reduced β1-integrin activity in WM852* cells (*Figure 5d–e*). The total β1-integrin was decreased about 20% in WM852* cells but this was not statistically significant (*Figure 5f–g*). On the other hand, neither *NOTCH3* depletion to 81% reduction in mRNA level altered the β1-integrin activation levels in WM852* (*Figure 5—figure supplement 2a–b*), nor, vice versa, inhibition of β1-integrin activation by the AIIB2 antibody treatment significantly altered *MMP14, NOTCH3* or its downstream targets *HEY1* and *HES1* at mRNA

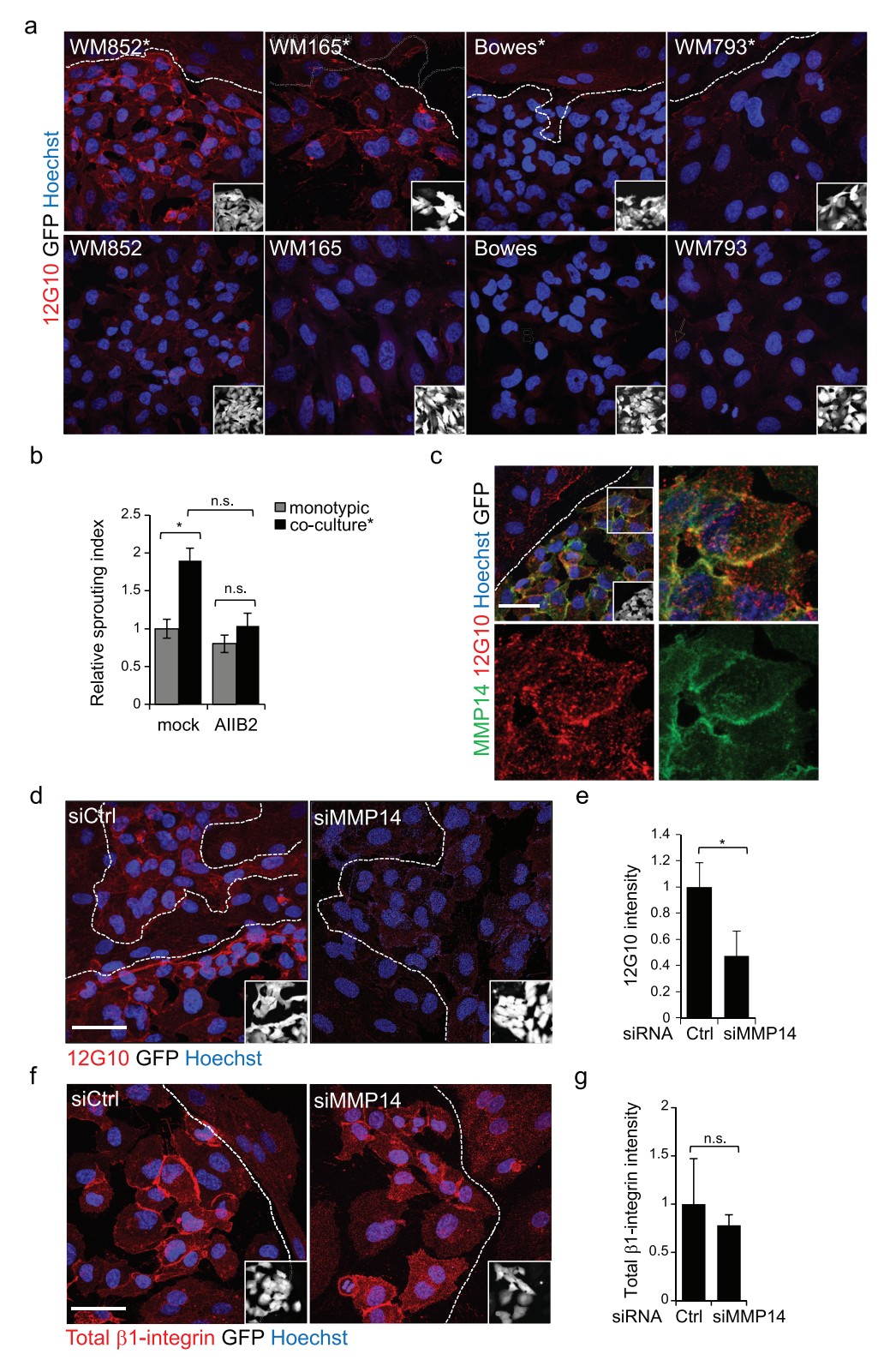

**Figure 5.** Change in the 3D growth phenotype of the LEC primed melanoma cells is β1-integrin dependent. (a) Representative confocal images of active β1-integrin (12G10) staining (red) in the indicated melanoma cell lines (GFP positive cells (white) in the inset) in the presence (*, upper panels) or absence (bottom panels) of LECs. Nuclei were counterstained with Hoechst 33342. The dashed line indicates the LEC-melanoma border. Scale bar = 50 μm. (b) Quantification of 3D sprouting index in WM852 and WM852* mock treated or treated with β1-integrin blocking antibody (AIIB2) during the 96

*Figure 5 continued on next page*

*Figure 5 continued*

hr fibrin growth assay. Graph shows the average of at least three images per condition per two independent experiments, error bars indicate the SEM; *: p<0.05. n.s., non-significant. (c) Representative confocal image of active β1-integrin (12G10, red) and MMP14 (green) staining of WM852* (white cells in the inset). Nuclei were counterstained with Hoechst 33342. The dashed lines indicate the border between LEC and WM852 (white, GFP positive WM852 cells in the inset). The right and bottom panels show an enlargement of the area enclosed within the white square as a merge, Notch3 (red) and MMP14 (green) in separated channels. Scale bar = 50 μm. (d,f) Representative confocal images of WM852* treated with the indicated siRNAs for 72 hr and stained for active β1-integrin with 12G10 (d, red), or total β1-integrin with P5D2 (f, red) antibodies. Nuclei were counterstained with Hoechst 33342. The dashed lines indicate the LEC-WM852 borders (white, GFP positive WM852 cells in the inset) border. Scale bar = 50 μm. Quantification of the average 12G10 (e) and total β1-integrin (g) signal intensity in WM852* (white) cells. Four images/condition were quantified from two independent experiments. More than 100 cells were always analysed per condition; error bars indicate SD. *: p<0.05. n.s., non-significant. Full size confocal images are available as a *source data 3*.

DOI: https://doi.org/10.7554/eLife.32490.013

The following figure supplements are available for figure 5:

**Figure supplement 1.** Quantification of active and total β1-integrin in monotypic and LEC-primed melanoma cells.

DOI: https://doi.org/10.7554/eLife.32490.014

**Figure supplement 2.** β1-integrin activity does not affect MMP14 and Notch3 expression.

DOI: https://doi.org/10.7554/eLife.32490.015

level (*Figure 5—figure supplement 2c*) or MMP14 and Notch3 protein levels as assessed by IFA (*Figure 5—figure supplement 2d–e*).

This suggests that the LEC-induced transient, invasively sprouting phenotype of melanoma cells is mediated by activated Notch3 and β1-integrin both dependent on the key upstream regulator MMP14.

## NICD3 ectopic expression is sufficient to induce 3D sprouting in non-metastatic WM793 cells

To further corroborate the role of MMP14 relocalization and activation of Notch3 and β1-integrin in the invasively sprouting phenotype induced by the LEC contact in the metastatic cell lines, we assessed if their expression or activation was able to induce this phenotype in the non-metastatic cell lines. First, we ectopically expressed NICD3, the constitutively active form of Notch3, and MMP14 in WM793 and Bowes cells. 24 hr after transfection the cells were subjected to the fibrin assay using vector-transfected cells as a negative control. Ectopic expression of NICD3 in WM793 induced the sprouting growth phenotype in fibrin resembling the growth phenotype of WM852* and WM165*, while the control WM793 cells continued to grow as sphere-like colonies (*Figure 6a*). However, ectopic expression of NICD3 in Bowes cells did not change their elongated, single cell invasive growth phenotype (data not shown). This confirms that in WM793, but not in Bowes cells, active Notch3 is sufficient to induce the invasively sprouting growth phenotype typical for the metastatic cell lines after the co-culture with LECs. When WM793 and Bowes cells were transfected with MMP14-expressing plasmid, no change was again seen in the 3D growth phenotype of Bowes cells, but surprisingly, ectopic expression of MMP14 induced cell death in WM793 as soon as 20 hr post-transfection (data not shown).

To address if β1-integrin activation was sufficient to induce the 3D sprouting phenotype in WM793, we cultured the cells on plates coated with the 12G10 β1-integrin activating antibody for 24 hr, and integrin activation was confirmed by 9EG7 antibody staining (*Figure 6b–c*). Despite β1-integrin activation no change in the 3D growth phenotype of WM793 in fibrin was observed, thus suggesting that activation of β1-integrin alone is not sufficient to induce the sprouting phenotype.

To further assess if the NICD3-induced sprouting phenotype in WM793 was dependent on MMP14 or β1-integrin activation, we treated the NICD3-transfected cells during the fibrin assay with the MMP14 inhibitor NSC405020 and β1-integrin blocking antibody AIIB2. While the NSC405020 treatment only mildly reduced the NICD3-mediated sprouting phenotype, AIIB2 treatment efficiently abolished it (*Figure 6d*).

Taken together, these experiments show that constitutive activation of Notch3 by NICD3, but not β1-integrin activation alone, can switch the phenotype of WM793 from a sphere-like to a invasively sprouting growth. Moreover, once Notch3 is constitutively activated, MMP14 inhibition has no additional effect on the sprouting, thus confirming the previous observation that MMP14 acts upstream

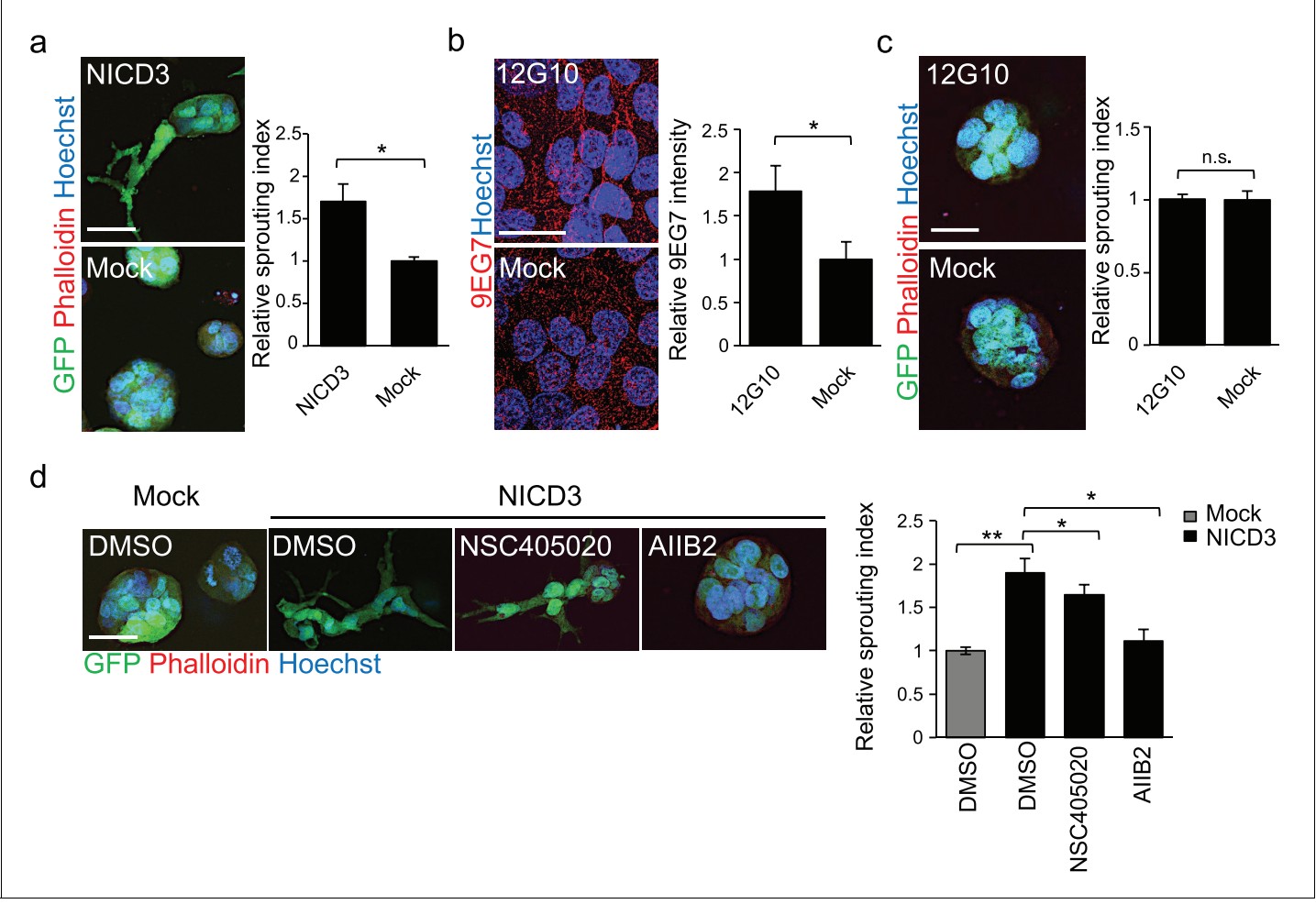

**Figure 6.** NICD3 overexpression provokes a β1 integrin-dependent 3D sprouting in WM793 cells. (a) Representative images (left) and quantification (right) of the 3D fibrin assay of WM793 upon transfection of either a NICD3 expressing vector or a control vector. Scale bar = 50 μm. Four images/condition were quantified from two independent experiments. Graph shows the average of at least three images per condition per two independent experiments, error bars indicate the SEM. *p<0.05. (b) Representative images (left) and quantification (right) as in of mock and 12G10 antibody treated WM793 cells. Active β1-integrin was detected with 9EG7 antibody (red). Nuclei were counterstained with Hoechst 33342. Four images/condition were quantified from two independent experiments. More than 100 cells were always analysed per condition; error bars indicate SD. Scale bar 50 μm. *p<0.05. (c) Representative images (left) and quantification (right) of 3D fibrin assay as in (a) of WM793 treated with 12G10 antibody for four days or mock treated. Scale bar = 50 μm. n.s., non-significant. (d) Representative images (left) and quantification (right) of 3D fibrin assay as in (a) of WM793 upon transfection of either a NICD3 expressing vector or a control vector (mock) and treated with either DMSO or the indicated inhibitors. Scale bar = 50 μm, *p<0.05; **p<0.01.

DOI: https://doi.org/10.7554/eLife.32490.016

of Notch3. Although the Notch3 induced phenotype was dependent on active β1-integrin (*Figure 6d*), Notch3 is not required for integrin activation (*Figure 5—figure supplement 2a–b*).

## In vivo invasion and dissemination of LEC-primed WM852 cells are dependent on MMP14 and Notch3

Human melanoma cells retain their invasive behaviour when transplanted to zebrafish embryos (*Chapman et al., 2014*). To further analyse the invasion capabilities of the LEC-primed WM852 cells in vivo, and the involvement of MMP14 and Notch3 in this process, we transplanted siRNA-treated WM852 and WM852* cells into pericardial cavity of zebrafish embryos and analysed the tumors four days later using intravital fluorescence microscopy (*Figure 7a and b*). In these experiments, LEC priming of WM852 prior to transplantation did not affect the size of primary tumors (*Figure 7c*), similarly to what was observed in the mouse xenograft studies (*Figure 1c*). However, co-culture of

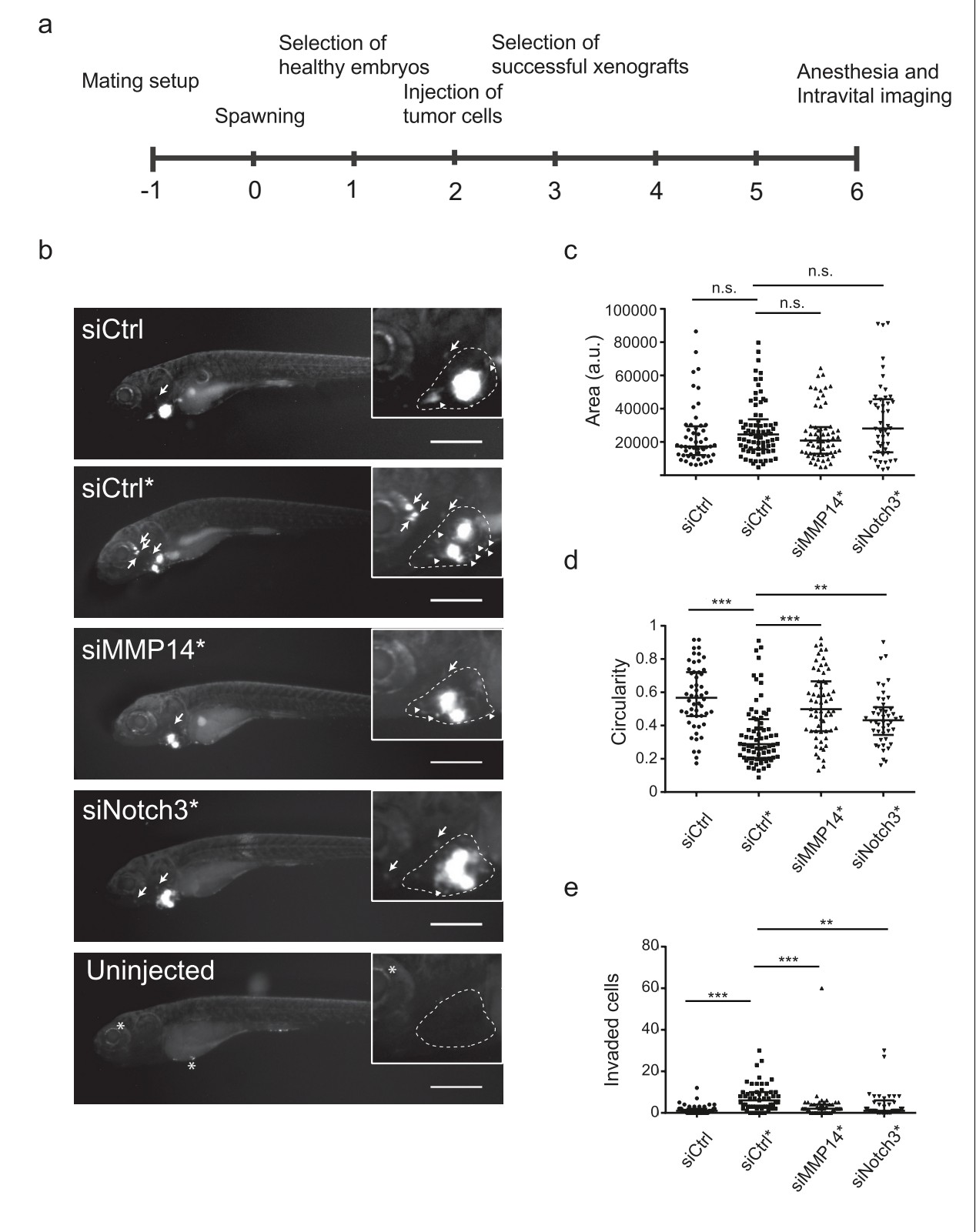

**Figure 7.** In vivo invasion and dissemination of LEC-primed melanoma cells are dependent on MMP14 and Notch3. (a) Time line of zebrafish xenograft experiments. (b) Intravital fluorescence microscopy images of six dpf zebrafish embryos taken four days post injection (4 dpi). Fluorescence in GFP channel is shown. Scale bar, 500 μm. Inset shows magnification of the primary tumor. Tumor cells invading outside pericardial space are marked with an arrow, invading cells in pericardial cavity with a triangle and unspecific fluorescence in eye and yolk sac with an asterisk (*). The outline of pericardial

*Figure 7 continued on next page*

*Figure 7 continued*

cavity is depicted with dashed line. (c) Quantification of area of primary tumors at 4 dpi. siCtrl, n = 52; siCtrl*, n = 74; siMMP14*, n = 61; siNotch3*, n = 46. (d) Quantification of circularity of primary tumors at 4 dpi. siCTRL, n = 52; siCtrl*, n = 74; siMMP14*, n = 61; siNotch3*, n = 46. (e) Quantification of melanoma cells invaded outside pericardial cavity. siCtrl, n = 43; siCtrl*, n = 55; siMMP14*, n = 44; siNotch3*, n = 37. (c–e) Non-parametric Kruskal-Wallis test with Dunn´s multiple comparison test was used, and in addition to individual data points, the median and interquartile range has been plotted. n.s., non significant (p>0.05); **p<0.01; ***p<0.001.

DOI: https://doi.org/10.7554/eLife.32490.017

WM852 cells with LECs did result in reduced circularity of the primary tumor (*Figure 7d*) indicating increased invasiveness in vivo. It also increased the number of cells that invaded outside the pericardial cavity and disseminated to distant parts in the embryo (*Figure 7e*). Importantly, depletion of either *MMP14* or *NOTCH3* in WM852* cells by siRNA reduced their invasiveness and dissemination in this model (*Figure 7d and e*), indicating that the LEC-induced increase in the metastatic behavior of WM852* cells is dependent on MMP14 and Notch3 in vivo.

## Discussion

Melanoma cells have been shown to be in close contact with the lymphatic vessels in human specimens and often metastasize via the lymphatic system, (*Niakosari et al., 2008*), implying that the melanoma cell-LEC interactions are common events in human melanomas. In addition, clinical data correlating metastatic spread with lymphatic infiltration suggest that melanoma-LEC interactions could contribute to the tumor progression and metastasis. Here we investigated the consequences of such interactions by coupling unique 2D and 3D LEC-melanoma co-culture models to in vivo mouse tumor model, transcriptome profiling, in vitro functional studies and a zebrafish xenograft/metastasis assay.

In the xenograft mouse model, dissemination of the metastatic WM852 cell line to distant organs was increased by LEC priming. Furthermore, in vitro molecular and functional studies revealed that the direct contact of LECs with metastatic melanoma cell lines triggered changes particularly in the localization of MMP14, an important protease for cancer-cell invasion (*Tatti et al., 2015*; *Turunen et al., 2017*), which in turn induced expression and activation of Notch3 and activation of β1-integrin leading to a significant change in the 3D growth phenotype of the melanoma cells. The interaction with LECs did not, however, induce metastasis or change the 3D growth phenotype of non-metastatic Bowes cells. Although Bowes have active MMP14 and an invasive, single cell 3D sprouting phenotype in vitro (*Tatti et al., 2011*), these intrinsic properties were not sufficient to support distant organ metastatic colonization in vivo. Moreover, ectopic expression of MMP14 or NICD3 did not induce any change in their 3D growth phenotype. Similar to Bowes, LEC priming did not markedly alter the 3D phenotype of another non-metastatic melanoma cell line WM793, that, despite of MMP14 expression (*Tatti et al., 2011*), continued to grow expansively in 3D as indicated by the sphere-like growth in 3D fibrin. However, these cells switched to the invasively sprouting growth upon introduction of NICD3, and the sprouting was dependent on β1-integrin. Another possible contributor to this change could be theβ1-integrin processing and trafficking, which appears quite different in Bowes as compared to WM793 and WM852. Based on our results, it is plausible that the capacity for expansive growth is a property required for the NICD3 dependent change in the 3D growth. Fittingly, WM852 cells with strong intrinsic capacity to grow expansively, switched to the invasively sprouting growth phenotype after the LEC-mediated transient activation of the MMP14-Notch3-β1-integrin axis, which resulted in increased MMP14 and Notch3 dependent metastasis in vivo. The LEC contact likewise switched the metastatic, expansively growing WM165 cells to invasive sprouting via activation of MMP14-Notch3-β1-integrin axis. Supporting the importance of this signaling axis, neither Bowes nor WM793 responded to the LEC contact by activating these key effectors needed for the metastatic phenotype.

Melanoma cell invasion into the deeper layers of the skin and distant sites is associated with molecular changes such as increased activation of the tissue degrading proteases including the MMPs (*Moro et al., 2014*; *Villanueva and Herlyn, 2008*). Our results demonstrate that the LEC contact increased MMP14 on the melanoma cell plasma membrane and cell-cell contacts in the metastatic melanoma cell lines. This was accompanied with an MMP14-dependent increase in Notch3

expression by a yet unidentified mechanism. MMP14 was recently shown to interact with and activate Notch1 at the cell membrane of melanoma cells, which supported melanoma cell growth (*Ma et al., 2014*). Although Notch receptors 1 and 3 are highly activated in melanoma with clear pro-tumorigenic functions, no Notch-activating mutations have been identified in melanoma. Therefore, the herein identified MMP14 dependent Notch3 upregulation may prove important in metastatic melanoma progression.

Our results also revealed that LEC contact induced β1-integrin activation in the metastatic WM852 and WM165 cells, which was required for the 3D sprouting of WM852* cells. β1-integrin activation was dependent on MMP14 as its depletion by siRNA abolished the β1-integrin activation and slightly, but non-significantly, reduced total β1-integrin protein levels These findings concur with previous studies demonstrating MMP14 incorporation into β1-integrin rich adhesion complexes (*Woskowicz et al., 2013*) and MMP14 binding to β1-integrin upon mammary branching morphogenesis (*Mori et al., 2013*) as well as its role in controlling the 3D cell-shape of stem cells through activation of β1-integrin signaling (*Tang et al., 2013*). Moreover, melanoma metastases express more activated β1-integrin than the primary tumors (*Kato et al., 2012*). Our current results add to this concept by demonstrating that endothelial cell contact activates integrins in melanoma cells, thus providing a putative mechanism for the contribution of the stromal lymphatic microenvironment to melanoma metastasis.

Negative regulator of β1-integrin, ICAP-1, has been shown to act as a potent inducer of Notch1-signaling by transcriptionally upregulating Notch ligands *DLL1* and *DLL4* and the downstream transcription factors *HEY1, HEY2,* and *HES5* in blood endothelial cells and thereby restrict the sprouting angiogenesis of the ECs (*Brütsch et al., 2010*). However, as our transcriptomic data of the LEC-primed melanoma cells did not reveal changes in *ICAP-1* expression upon LEC-priming it probably does not play a significant role in the Notch3-dependent invasive sprouting of metastatic melanoma cells. Furthermore, our results indicate that β1-integrin functions downstream of Notch3 in the LEC-primed melanomas.

Although melanoma cells at the primary tumor site predominantly disseminate through the lymphatic vascular route, the distant organ metastasis requires cells entering blood circulation and colonization in the distant organ tissue. MMP14, Notch3 and β1-integrin are all proteins linked to epithelial-to-mesenchymal transition in carcinoma cell metastasis (*Cao et al., 2008*; *Kato et al., 2012*; *Liu et al., 2014*), supporting the idea that the metastatic WM852* and WM165* melanoma cells have also undergone a transient transition into a more mesenchymally invasive state that can promote hematogenous dissemination. However, since the Bowes cells with high MMP14 levels did not metastasize, MMP14 activity by itself is apparently not enough for efficient distant organ metastasis. In the absence of strong cancer cell-cell interactions, constitutively high MMP14 activity can facilitate single cell invasion, thereby compromising the adhesive and expansive growth potential essential for efficient tissue colonization of the tumor cells. We therefore propose that the capacity for transiently induced 3D invasive sprouting coupled to expansive growth behavior are required for the most aggressive melanoma cells to enter and survive in the blood circulation, as well as to efficiently colonize the distant organs.

Melanoma progression is suggested to be driven by reversible and functional reprogramming of different signaling routes, known as reversible phenotypic plasticity of cell (*Vandamme and Berx, 2014*). Changes and interactions in the tumor microenvironment are believed to direct this phenotype-switching, but the detailed mechanisms are largely unknown. Our current results support the role of reversible phenotype-switching in melanoma progression and identifies the stromal lymphatic endothelium as one of the key triggers for the process to support both hematogenous dissemination and tissue colonization at the distant organs.

# Materials and methods

**Key resources table**

| Reagent type (species) or resource | Designation | Sourcor reference | Identifiers | Additional Information |
|---|---|---|---|---|

*Continued on next page*

*Continued*

| Reagent type (species) or resource | Designation | Sourcor reference | Identifiers | Additional Information |
|---|---|---|---|---|
| Danio Rerio casper strain (*roy-/-; mitfa-/-*) | | PMID: 18371439 | | |
| Mus Musculus C.B-17/IcrHanTMHSD | | Harlan, Indianapolis, IN, USA | | |
| Human primary juvenile foreskin lymphatic endothelial cells | | Promocell, Heidelberg, Germany | | |
| Adult Dermal lymphatic endothelial cells | | Lonza, Basel, Switzerland | | |
| WM852 | | Wistar Institute Philadelphia | RRID:CVCL_6804 | |
| WM165 | | Wistar Institute Philadelphia | RRID:CVCL_L033 | |
| Bowes | | D.B. Rifkin, Rockfeller University | RRID:CVCL_3317 | |
| WM793 | | Wistar Institute Philadelphia | RRID:CVCL_8787 | |
| HEK293FT | | Biomedicum Functional Genomic Unit, University of Helsinki | | |
| siRNA | Notch3 siRNA | Dharmacon, Lafayette, CO | L-011093-00-0005 | |
| | Notch3 siRNA | Ambion, Waltham, MA | 4392420 | |
| siRNA | MMP14 siRNA | Qiagen, Hilden, Germany | SI03648841; SI00071176 | |
| Transfected construct | NICD3pCLE | AddGene, Cambridge, MA; PMID: 16508304 | Plasmid #26894 | |
| Transfected construct | MMP14 Sport6 | GenomeBiology Unit University of Helsinki | NA | |
| Transfected construct | PcDNA3 | Invitrogen, Carlsbad, CA | NA | |
| Antibody | Pecam | DAKO, Santa Clara, CA | M0823 | IFA 3D 1:500 |
| Antibody | VE-cadherin | BD Pharmingen, San Jose, CA | 553927 | IFA 1:1000 |
| Antibody | GFP | Prof. Jason Mercer, UCL, London | NA | IFA 3D 1:1000 |
| Antibody | Notch3 | Santa Cruz Biotechnology, Dallas, TX | sc-5593 | IFA 1:50 WB 1:250 |
| Antibody | MMP14 EP1264Y | Abcam, Cambridge, UK | 51074 | IFA 1:100 |
| Antibody | MMP14 (LEM clone) | Chemicom, Waltham, MA | MAB3328 | IFA 1:300 FACS 1:100 |
| Antibody | active β1-integrin 12G10 | Abcam | 30394 | IFA 1:300 FACS 1:100 |
| Antibody | active β1-integrin 9EG7 | BD Pharmingen | 553715 | IFA 1:100 FACS 1:100 |
| Antibody | Total β1-integrin | Abcam | 52971 | WB 1:1000 |
| Antibody | TGN46 | Sigma, St. Louis, MO | T7576 | IFA 1:500 |
| Antibody | LYVE1 | Reliatech, Wolfenbüttel, Germany | 103-PA50AG | IHC 1:200 |
| Antibody | Alexa594-phalloidin | Thermo Fisher, Waltham, MA | 21833 | IFA 3D 1:200 |
| Sequence-based reagents | | | | |
| CD31 RTqPCR primers (for, rev) | AACAGTGTTGACATGAAGAGCC, TGTAAAACAGCACGTCATCCTT | | | |
| CD34 RTqPCR primers (for, rev) | TGGGCATCACTGGCTATTTC, CCACGTGTTGTCTTGCTGAA | | | |
| FLT4 RTqPCR primers (for, rev) | GACAGCTACAAATACGAGCATCTG, CTGTCTTGCAGTCGAGCAGAA | | | |

*Continued on next page*

Continued

| Reagent type (species) or resource | Designation | Sourcor reference | Identifiers | Additional Information |
|---|---|---|---|---|
| NOTCH1 RTq PCR primers (for, rev) | GAGGCGTGGCAGACTATCATGC, CTTGTACTCCGTCAGCGTGA | | | |
| NOTCH2 RTq PCR primers (for, rev) | CCTGGGCTATACTGGGAGCTACTG ,ACACCCTGATAGCCTGGGACAC | | | |
| NOTCH4 RTq PCR primers (for, rev) | AATCCCACTGCCTCCAGACT, TTGTGGCAAAGGGAAGAGAC | | | |
| HES1 RTq PCR primers (for, rev) | TCAACACGACACCGGATAAA, TCAGCTGGCTCAGACTTTCA | | | |
| HEY1 RTq PCR primers (for, rev) | GTTCGGCTCTAGGTTCCATGT, CGTCGGCGCTTCTCAATTATTC | | | |
| HEY2 RTq PCR primers (for, rev) | TTGAGAAGACTTGTGCCAACTG, GTGCGTCAAAGTAGCCTTTACC | | | |
| MMP14 RTq PCR primers (for, rev) | GCAGAAGTTTTACGGCTTGCAA, CCTTCGAACATTGGCCTTGAT | | | |
| ACT RTq PCR primers (for, rev) | TCACCCACACTGTGCCATCTACGA, CAGCGGAACCGCTCATTGCCAATGG | | | |
| GAPDH RTqPCR primers (for, rev) | TCACCACCATGGAGAAGGCT, GCCATCCACAGTCTTCTGGG | | | |
| Commercial assay or kit | NucleoSpin RNA II kit | Macherey Nagel,Düren, Germany | 740955 | |
| Commercial assay or kit | SYBR Green PCR mix | Fermentas, Waltham, MA | 4415440 | |
| Commercial assay or kit | QuantiTect Primer Assay NOTCH3 | Qiagen | QT00003374 | |
| Commercial assay or kit | dextran coated magnetic nanoparticles extran coated magnetic nanoparticles | fluidMAG-DX, Chemicell, Berlin, Germany | 4104–5 | |
| Commercial assay or kit | MidiMACS separator | Miltenyi Biotec,Bergisch Gladbach, Germany | 130-042-302 | |
| Commercial assay or kit | LS column | Miltenyi Biotec | 130-042-401 | |
| Chemical compound, drug | Lipofectamine RNAiMax | Invitrogen | 13778150 | |
| Chemical compound, drug | DAPT | Sigma | D5942 | Used at 10 μM |
| Chemical compound, drug | GM6001 | Tocris Biosciences, Bristol, UK | 2983/10 | Used at 10 μM |
| Chemical compound, drug | NSC 405020 | Selleckchem,Munich, Germany | S8072 | Used at 50 μM |
| Chemical compound, drug | AIIB2 | DSHB hybridoma, from Johanna Ivaska | RRID:AB_528306 | Used 1:10 |
| Software, algorithm | Bioimage XD (http://www.bioimagexd.net/) | PMID: 22743773 | NA | |
| Software, algorithm | CellProfiler | PMID: 17076895 | RRID:SCR_007358 | |
| Software, algorithm | Adobe Photoshop | | RRID:SCR_014199 | |
| Software, algorithm | ImageJ | | RRID:SCR_003070 | |
| Software, algorithm | Pathview https://pathview.uncc.edu | | RRID: SCR_002732 | |
| Software, algorithm | Morpheus https://software.broadinstitute.org/morpheus/ | | RRID: SCR_014975 | |

## Cell lines

Human primary juvenile foreskin lymphatic endothelial cells (LEC) were obtained from Promocell, and adult dermal LECs from Lonza. They were cultured in endothelial cell culture media (EBM-2, Lonza) supplemented with the growth factors provided (except VEGF) and 5% fetal calf serum (full media referred as EGM-2). The human melanoma cell lines WM852, WM165 and WM793 (Wistar Institute, Philadephia, PA) and Bowes (a kind gift from Dr. D. B. Rifkin, Rockefeller University, New York, USA), as well as HEK293-FT cells (obtained from Biomedicum Functional Genomics Unit, FuGU) used for lentivirus production were cultured in Dulbecco's Modified Eagle Medium as previously described (*Tatti et al., 2011*). In most experiments, the melanoma cells were either traced with Vybrant CFDA SE cell tracer (Invitrogen), or transduced with retroviruses expressing a dual eGFP-luc-reporter as described in (*Tatti et al., 2011*). The cell lines have been regularly tested negative for mycoplasma. The source of the cell lines is also reported above in the key resources table in Materials and methods

## 3D fibrin assays

To study the interaction of melanoma cells with the LECs in 3D, they were embedded into a cross-linked fibrin (Calbiochem, San Diego, CA) matrix together with preformed LEC spheroids and cultured in the endothelial medium for 72 hr. The assay was adapted from a previously reported angiogenesis assay (*Korff and Augustin, 1998*) essentially as described in *Tatti et al. (2015)*. To study the invasive potential of the separated melanoma cells after the LEC co-culture, the melanoma cells were embedded into the fibrin matrix as single cells and allowed to grow for four days as described in (*Tatti et al., 2011*). The melanoma invasion into the EC spheroids as well as to the fibrin matrix was analysed by immunofluorescent stainings, and confocal microscopy.

## 2D co-culture and cell separation

For the 2D co-culture, the LECs were seeded together with the melanoma cells on gelatin or fibronectin (Sigma) pre-coated cell culture plates in a 2:1-4:3 ratio in EGM-2 media. The melanoma cells cultured in EGM-2 showed no signs of compromised cell survival. The co-cultures were grown for 24–72 hr prior further use for immunofluorescent stainings or separations. For the separations, the melanoma cells were loaded with dextran coated magnetic nanoparticles (1 mg/ml, fluidMAG-DX, Chemicell) for 24–48 hr. The nanoparticle-containing melanoma cells were separated from the LECs using the MidiMACS separator and LS column (both from Miltenyi Biotec), after which the cell suspensions were used for functional assays, or lysed for RNA extraction. To study the paracrine effects, the supernatants from melanoma/LEC/co-cultures were collected after 48 hr of culture, filtered, and applied onto the melanoma cells for 48 hr, after which the melanoma cells were analysed by immunofluorescent stainings or qRT-PCR.

## RNA sequencing (RNA-seq)

RNA extraction for the RNA sequencing analysis was done from three independent experiments with a TRI reagent (Sigma) protocol supplemented with phenol-chloroform precipitation step. The RNA concentrations were measured with NanoDrop, and Bioanalyzer (Agilent Technologies, Santa Clara, CA) analysis was performed to check the RNA quality. RNA sequencing was performed with NextSeq500 sequencer (Illumina, San Diego, CA ) as quadruplicates. The data was aligned to HS GRCh38.76 reference genome, and the differentially expressed genes were obtained by using DESeq2 Bioconductor package (*Love et al., 2014*). Non-expressed genes (average under five counts/sample) and ribosomal RNAs were filtered out. Genes with adjusted p-values less than 0.05 were considered significant. Generally applicable gene set enrichment (GAGE) Bioconductor package was used for pathway analysis, and KEGG pathway maps were rendered with Pathview (https://pathview.uncc.edu). Morpheus (https://software.broadinstitute.org/morpheus/) was used to generate the gene heatmap. Individual gene/transcript expressions are shown as FKPM (fragments per kilobase of exon per million fragments mapped) values. The RNA-Seq data is deposited in NCBI GEO, with accession number GSE100269 (https://www.ncbi.nlm.nih.gov/geo/query/acc.cgi?acc=GSE100269).

## Real time quantitative PCR (qRT-PCR)

RNA was isolated using the NucleoSpin RNA II kit (Macherey Nagel) and the transcripts were measured by qRT-PCR as previously described (*Cheng et al., 2011*). Briefly, the Lightcycler 480 (Roche, Basel, Switzerland) qRT-PCR system was used, and the reactions were done using the SYBR Green PCR mix (Fermentas) and QuantiTect Primer Assay against *NOTCH3* (QT00003374, Qiagen). *GAPDH* or *ACT* were used as endogenous controls.

## RNA interference

Cell monolayers cultured in 96-, 24- or 6-well plates were treated with siRNA according to manufacturers' instructions. All siRNAs were used at a final concentration of 10–25 nM and cells transfected using lipofectamine RNAiMax (Invitrogen) for 24–72 hr. The following siRNAs were used: unspecific control (Ambion,Waltham, MA4390843; Dharmacon, D-001810-10-05; Qiagen 1027281), Notch3 (Ambion, 4392420; Dharmacon, L-011093-00-0005), MMP14 (Qiagen, SI03648841; SI00071176).

## Inhibitor treatments

Gamma-secretase inhibitor DAPT (Sigma) and pan MMP inhibitor GM6001 (Tocris Biosciences) at 10 µM concentrations, MMP14 hemopexin domain inhibitor NSC 405020 (Selleckchem) at 50 µM and the β1-integrin blocking antibody AIIB2 were applied to the growth medium during the 48 hr LEC-WM852 2D co-cultures and also to the 96 hr 3D fibrin assays when indicated.

## Plasmids and transient transfection

Bowes and WM793 were plated in a 24 well plate one day prior transfection to reach 80–90% confluency next day. Cells were then transfected with 3 µg of one of the following plasmids Sport6-MMP14 (Genome Biology Unit, University of Helsinki), NICD3-pCLE (addGene) or pcDNA3 as control vector using Lipofectamine 2000 (Thermo Fisher Scientific, Waltham, MA) accoding to manufacturer´s instruction. One day post transfection cells were used for the appropriate experiment.

## Indirect immunofluorescence and imaging

The 2D cultured and 3D fibrin cultured cells as well as adhesion assays were stained as previously described (*Cheng et al., 2011*) with antibodies against: PECAM (Dako, M0823), VE-cadherin (BD Pharmingen, 553927), GFP (a kind from Prof. Jason Mercer, UCL, London), Notch3 (Santa Cruz, sc-5593), MMP14 (Abcam, 51074), active β1-integrin (12G10, Abcam, 30394; 9EG7, BD Pharmingen, 553715) and total β1-integrin (P5D2, DSHB hybridoma; from Johanna Ivaska). The secondary antibodies conjugated with Alexa488, Alexa594 and Alexa647 fluorochromes were used to visualize the stainings, and Alexa594 conjugated phalloidin (Invitrogen) was used to stain the actin filaments. The nuclei were counterstained with Hoechst 33342. The fluorescent images were acquired using a Zeiss epifluorescent microscopes, Cellinsight automated epifluorescent microscope (Thermo Scientific), and a Zeiss LSM780 or Leica SP5 confocal imaging systems.

## Western blot

Melanoma cells were lysed in RIPA buffer containing protease and phosphatase inhibitor cocktails (Thermo Scientific) and protein concentration was obtained using Bio-Rad protein assay dye reagent concentrate (Bio-Rad, Hercules, CA). Equal amounts of proteins were loaded on 4–15% SDS-PAGE gel (Bio-RAD). SDS PAGE was run at 55mA for 50 min and proteins were transferred on nitrocellulose membrane (Bio-RAD). The blots were blocked for 45 min in 5% non-fat dry milk and probed using rabbit anti-Notch3 (Santa Cruz) or rabbit anti-ITGB1 (Abcam, 553715). Mouse anti-β-actin (Sigma) antibody was used as loading control. Primary antibodies were incubated 1 hr at room temperature followed by incubation in HRP-conjugated secondary antibody for 1 hr at room temperature (goat-anti mouse IgG and goat anti-rabbit IgG, Millipore, Burlington, MA). Bands were detected by chemiluminescence using ECL solution (WesternBright Sirius, Advansta, Menlo Park, CA) and visualized by Chemi-Doc (Bio-Rad).

## Flow cytometry

Cells were detached using hyclone HyQtase (Thermo Fisher), washed once with full medium and once with PBS. Cells were then fixed in 2%PFA for 15 min RT and washed in tyrodes buffer (10 mM

Hepes-NaOH pH7.5, 137 mM NaCl, 2,68 mM $NaH_2PO_4$, 1,7 mM $MgCl_2$, 11.9 mM $NaHCO_3$, 5 mM glucose, 0.1%BSA). Subsequently $1.5 \times 10^5$ cells / condition were stained with anti-MMP14 (Chemicon, MAB3328) antibody for 1 hr at 4°C under rotation, washed once in tyrodes buffer and incubated with Alexa647-conjugated secondary antibody for 1 hr at 4°C under rotation. Cells were then washed once in tyrodes buffer and GFP positive (melanoma) cells were analysed for Alexa647 intensity.

## In vivo tumorigenicity and metastasis assay

To study the tumorigenicity and metastatic capacity of the melanoma cells from monotypic cultures or after LEC priming, the GFP-luc reporter containing melanoma cells were first cultured with or without the LECs in the 3D fibrin matrix as described above. After 72 hr of co-culture, the proteinase inhibitor approtinin was removed from the culture media and melanoma cells were allowed to digest the matrix. After 48–72 hr, the cells were collected, and $1 \times 10^6$ cells were injected subcutaneously into C.B-17/IcrHanTMHSD-Prkdc Scid mice (Harlan). The mice and tumor size were followed weekly for up to 70 days, after which the tumor volumes and weights were also measured. After the follow-up period, mice were sacrificed, the tumors were collected for further analyses by immunohistochemical stainings, and the metastasis was analysed by measuring the luciferase activity in the isolated organs using the Caliper IVIS Kinetic imaging system. In addition, the metastatic human cells were detected from mouse lung genomic DNA by quantitative PCR (q-PCR) against human Alu sequences, and using mouse genomic DNA as normalization as previously described (*Liu et al., 2011*).

## Zebrafish xenograft and metastasis assay

A detailed description of the experimental procedure for this assay is provided at Bio-protocol (*Paatero et al., 2018*). Adult zebrafish (Danio Rerio) of casper strain (*roy-/-; mitfa-/-*) (*White et al., 2008*) were maintained according to standard procedures (*Nuesslein-Volhard and Dahm, 2011*; *Westerfield and Zon, 2009*) and embryos were collected after natural spawning. Experimentation with zebrafish was performed under licence ESAVI/9339/04.10.07/2016. The zebrafish embryos were cultured in E3-medium (5 mM NaCl, 0.17 mM KCl, 0.33 mM $CaCl_2$, 0.33 mM $MgSO_4$) supplemented with 0.2 mM phenylthiourea (PTU, Sigma-Aldrich) at 33°C. Two days post-fertilization, the embryos were anesthesized with MS-222 (200 mg/l, Sigma-Aldrich) and mounted into low-melting point agarose for tumor transplantation. Prior to transplantation, the co-cultured and siRNA-treated WM852-GFP melanoma cells were prepared and separated from LECs as described above. Approximately 5–10 nl of melanoma cell suspension was microinjected into pericardial cavity of the embryo using CellTramVario (Eppendorf), Injectman Ni2 (Eppendorf) micromanipulator and borosilicate glass needles pulled from glass capillaries (TW100-4, World Precision Instruments Ltd., Sarasota, FL) using micropipette puller (PB-7, Narishige, Tokyo, Japan). After transplantation, the embryos were released from the agarose and cultured in E3-PTU at 33°C. On the following day, the successfully xenografed healthy embryos were selected to the experiment and placed into 12-well plates (1 embryo per well). At 6 dpf (4 days post-injection) the embryos were anaesthesized with MS-222 and imaged in lateral orientation with Zeiss StereoLumar V12 fluorescence microscope.

The circularity and area of the primary tumor was measured manually using FIJI software (ImageJ version 1.49 k) (*Schindelin et al., 2012*). In cases where embryo carried more than one primary tumor, the largest nodule was considered as primary tumor and measured, or in cases where equally sized nodules existed, all of them were measured. The number of invaded cells were counted manually based on GFP-fluorescence. Only invading cells outside the pericardial cavity were counted. Invading cells above the yolk sac or in the lens were not also counted as these sites tend to have prominent autofluorescence. Samples having significant malformations and images where embryo was not laterally oriented were excluded from the analysis. Samples were not blinded for imaging and subsequent analyses.

## Immunohistochemistry

The mouse tumors were stained with antibodies against MMP14 (Chemicon), Lyve-1 (Reliatec) and Notch3 (Santa Cruz) as previously described (*Cheng et al., 2011*). The antibody stainings were visualized using Alexa594 and Alexa647 fluorochrome conjugated secondary antibodies for MMP14 and

LYVE-1 and anti-rabbit HRP and DAB as a substrate for detection of Notch3. The images were acquired with 3DHistech Panoramic 250 FLASH II digital slide scanner or Zeiss LSM780 confocal imaging system.

Notch3 staining was scored as low, medium, or high according to intensity. Scoring was performed by three independent investigators (SA, SG, PMO) without knowledge of the sample origin. Differing scores were discussed and consensus scores were determined.

## Statistical analysis

For quantification of the assay in 3D-fibrin gels, confocal stacks where analysed with the open source software Bioimage XD (http://www.bioimagexd.net/) (*Kankaanpää et al., 2012*) using the 'skeleton' tool in the 3D modules of the software. Cell clusters were identified by thresholding the intensity of melanoma GFP fluorescence. The skeleton function of Bioimage XD shrunk the 3D object from all directions until a central segment (the skeleton of the object) was obtained. The sum of the length of all the segments within a 3D object were considered as a measure of sprouting of the melanoma cells in 3D. This value was calculated for all objects in each image (typically around 50 objects per image). The mean and SD of all these values from at least three images per condition was given as the final value of sprouting in each experiment.

For quantification of the intensity of fluorescent stainings, mean intensity and respective SD in melanoma (GFP expressing) cells was measured using CellProfiler pipeline. At least four images were quantified in each different condition. Experiments were repeated at least two times, p-values was calculated with a one-tailed unpaired Student's t-test. *$p < 0.05$, **$p < 0.01$.

For quantification of western blotting, band intensities were measured in two independent experiments using Image Lab quantification program. For each sample, the intensities were first normalized to the corresponding loading control, then the average intensity was calculated. The mean intensity and SD were calculated from two experiments.

For statistical analysis of the qRT-PCR data, logarithmic values were converted to ddCt values (linear log2 scale values) and p-values were calculated with a one-tailed unpaired Student's t test. *$p < 0.05$, **$p < 0.01$, ***$p < 0.001$.

Non-parametric Kruskal-Wallis test with Dunn's multiple comparison test (GraphPad Prism 6.05, GraphPad Software, La Jolla California USA,) was used in the analyses of zebrafish data. Each condition was compared siCTRL-LEC co-culture. In each graph, median and interquartile range has been plotted. *$p < 0.05$, **$p < 0.01$, ***$p < 0.001$.

## Acknowledgements

We thank Veronika Rezov, Jenny Bärlund and Sari Tynkkynen for excellent technical assistance, and Kari Alitalo for critical comments and provided reagents. The DNA Sequencing and Genomics Laboratory at the Institute of Biotechnology is granted for performing the RNA sequencing. Genome Biology Unit, Biomedicum Imaging Unit, Light Microscopy Unit (University of Helsinki) for providing expert service on imaging. This work was supported by the Centre of Excellence grants from the Academy of Finland (Translational Cancer Biology; PMO, PS, JI) and (Cancer Genetics Research; SH). Finnish Cancer Foundations (PMO, PP, KL), Sigrid Juselius Foundation (PMO), and University of Helsinki Foundations (PMO). PP was supported by the Helsinki Biomedical Graduate Program (HBGP; University of Helsinki), the Finnish Medical Foundation, and Emil Aaltonen Foundation. SA was supported by the Doctoral Program in Biomedicine (DPBM; University of Helsinki). GB was supported by the Foundation for the Finnish Cancer Institute.

## Additional information

### Funding

| Funder | Grant reference number | Author |
| --- | --- | --- |
| Terveyden Tutkimuksen Toimikunta | 307366 | Johanna Ivaska Pipsa Saharinen Päivi M Ojala |

| Finnish Cancer Foundation | | Johanna Ivaska<br>Pipsa Saharinen<br>Sampsa Hautaniemi<br>Päivi M Ojala |
| --- | --- | --- |
| Sigrid Juséliuksen Säätiö | | Kaisa Lehti<br>Päivi M Ojala |
| University of Helsinki | Doctoral program in Biomedicine | Pirita Pekkonen<br>Sanni Alve |
| University of Helsinki Foundation | | Päivi M Ojala |
| Suomen Lääketieteen Säätiö | | Pirita Pekkonen |
| Emil Aaltosen Säätiö | | Pirita Pekkonen |
| Terveyden Tutkimuksen Toimikunta | 309544 | Silvia Gramolelli |
| Terveyden Tutkimuksen Toimikunta | | Sampsa Hautaniemi |

The funders had no role in study design, data collection and interpretation, or the decision to submit the work for publication.

## Author contributions

Pirita Pekkonen, Conceptualization, Resources, Formal analysis, Supervision, Funding acquisition, Investigation, Methodology, Writing—original draft, Project administration, Writing—review and editing; Sanni Alve, Conceptualization, Formal analysis, Investigation, Methodology, Writing—original draft; Giuseppe Balistreri, Formal analysis, Investigation, Methodology; Silvia Gramolelli, Olga Tatti-Bugaeva, Conceptualization, Formal analysis, Supervision, Investigation, Methodology, Writing—original draft, Writing—review and editing; Ilkka Paatero, Data curation, Formal analysis, Supervision, Investigation, Methodology, Writing—review and editing; Otso Niiranen, Data curation, Formal analysis, Methodology; Krista Tuohinto, Nina Perälä, Nadezhda Zinovkina, Pauliina Repo, Investigation, Methodology; Adewale Taiwo, Methodology, acquisition of data; Katherine Icay, Data curation, Methodology; Johanna Ivaska, Pipsa Saharinen, Formal analysis, Methodology, Writing—review and editing; Sampsa Hautaniemi, Resources, Methodology; Kaisa Lehti, Conceptualization, Formal analysis, Investigation, Methodology, Writing—original draft, Writing—review and editing; Päivi M Ojala, Conceptualization, Resources, Data curation, Formal analysis, Supervision, Funding acquisition, Investigation, Writing—original draft, Project administration, Writing—review and editing

## Author ORCIDs

Johanna Ivaska https://orcid.org/0000-0002-6295-6556
Kaisa Lehti https://orcid.org/0000-0001-9110-8719
Päivi M Ojala https://orcid.org/0000-0001-9065-1832

## Ethics

Animal experimentation: All of the animals were handled according to approved institutional animal care and use committee protocols of the University of Helsinki (. The protocol was approved by the National Animal Experiment Board (Eläinkoelautakunta ELLA; Permit Number: ESAVI/434/04.10.03/2012). Experimentation with zebrafish was approved by the National Animal Experiment Board and performed under licence ESAVI/9339/04.10.07/2016.

## Decision letter and Author response

Decision letter https://doi.org/10.7554/eLife.32490.031
Author response https://doi.org/10.7554/eLife.32490.032

## Additional files

### Supplementary files

• Soure data 1. Full size confocal images of *Figure 2*.
DOI: https://doi.org/10.7554/eLife.32490.018

• Soure data 2. Full size confocal images of *Figure 4a,c,e*; *Figure 4—figure supplement 1b,c*; *Figure 4—figure supplement 2a,c*.
DOI: https://doi.org/10.7554/eLife.32490.019

• Soure data 3. Full size confocal images of *Figure 5a,c,d,f*.
DOI: https://doi.org/10.7554/eLife.32490.020

• Soure data 4. Full size confocal images of *Figure 5—figure supplement 1b,c*; *Figure 5—figure supplement 2a,b,d,e*.
DOI: https://doi.org/10.7554/eLife.32490.021

• Supplementary file 1. related to *Figure 2* and containing more than two fold significantly altered genes.
DOI: https://doi.org/10.7554/eLife.32490.022

• Supplementary file 2. related to *Figure 2* and containing all genes significantly up- and downregulated listed for both cell lines in the selected pathways.
DOI: https://doi.org/10.7554/eLife.32490.023

• Transparent reporting form
DOI: https://doi.org/10.7554/eLife.32490.024

### Major datasets

The following dataset was generated:

| Author(s) | Year | Dataset title | Dataset URL | Database, license, and accessibility information |
|---|---|---|---|---|
| Ojala P, Niiranen O | 2017 | LEC-induced gene expression changes in melanoma cells | https://www.ncbi.nlm.nih.gov/geo/query/acc.cgi?acc=GSE100269 | Publicaly available at the NCBI Gene Expression Omnibus (accession no:GSE100269) |

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
