## [Decision Letter]

Thank you for submitting your article "Lymphatic endothelium stimulates melanoma metastasis and invasion via MMP14-dependent Notch3 and β1-integrin activation" to *eLife* for consideration as a Feature Article. Your article has been favorably evaluated by Fiona Watt (Senior Editor) and three reviewers, one of whom, Reinhard Fässler (Reviewer #1), is a member of our Board of Reviewing Editors. The following individual involved in review of your submission has agreed to reveal their identity: Corinne Albiges-Rizo (Reviewer #3).

The reviewers have discussed the reviews with one another and the Reviewing Editor has drafted this decision to help you prepare a revised submission.

Summary:

Pekkonen et al. show that lymphatic endothelial cells induce a metastatic program by sequentially activating MMP14, Notch3 and β1 integrin signaling in melanoma cells, which in turn promotes their invasion in vitro and metastasis in vivo.

Essential revisions:

1) The link between MMP14, Notch3 and β1 integrin is not worked out well. Please test whether Notch3 depletion and/or inhibition alter MMP14 levels and β1 integrin activation, and whether inhibition of β1 integrins changes MMP14 and Notch3 levels and Notch target gene expression.

2) MMP14 and β1 integrin levels should be quantified by FACS. Immunostaining is not quantitative and furthermore, shows an unclear perinuclear accumulation of MMP14. What does the perinuclear location mean? The corresponding Western blot for Figure 5—figure supplement 1C should be shown, and β1 integrin activation verified with the rat mAb 9EG7.

3) The dataset derived from a comparison of melanoma cell lines with and without LEC priming is interesting and the list(s) of genes with altered expression should be shown as supplementary data. Furthermore, analyze and describe in a few sentences (or an additional figure panel) the genes and/or pathways altered in both cell lines after LEC priming and the genes specific for WM852 or Bowes cells.

4) To confirm validity of the pathway induced by LECs, test whether the sprouting phenotype is induced in Bowes and WM793 cells by upregulating MMP14 on the cell surface or by activating Notch3 or β1 integrin signaling.

5) Verify the involvement of Notch3 and MMP14 in the in vivo metastatic dissemination of 3D LEC-co-cultured melanoma cells.

6) The LEC/tumor cell interaction should be better characterized with the available transcriptomic data to potentially identify a cadherin switch, adhesion receptor switch, homo or heterotypic interaction, needs of specific ECM, etc.

The text says that "global gene expression revealed change in ECM-receptor interaction". What is meant with this sentence?

[Editors' note: further revisions were requested prior to acceptance, as described below.]

Thank you for resubmitting your work entitled "Lymphatic endothelium stimulates melanoma metastasis and invasion via MMP14-dependent Notch3 and β1-integrin activation" for further consideration at *eLife*. Your article has been favorably evaluated by Fiona Watt (Senior Editor) and three reviewers, one of whom, Reinhard Fässler, is a member of our Board of Reviewing Editors.

The manuscript has been improved but there are some remaining issues that need to be addressed, as outlined below:

It is shown that the LEC-induced metastatic melanoma 3D growth phenotype is Notch3- and β1 integrin-dependent. Despite the evidence showing that MMP14 is upstream of Notch3 and β1 integrin, the authors do not provide convincing experimental data for a MMP14-Notch3- β1 integrin axis, as claimed in the second paragraph of the Discussion. It is demonstrated that β1 integrin is needed for dissemination once Notch3 is activated. However, evidence for a direct effect of Notch3 on β1 integrin activity is missing. Instead, the findings shown in the manuscript suggest that MMP14 acts separately on Notch3 and β1 integrin. This issue requires clarification. In addition, the statistics are incomplete and the quality of some images requires improvement.

1) Statistics have to be provided for all graphs. This is essential.

2) Figure 2—figure supplement 2: the lack of significance in panel G, which shows an increase of Notch3 mRNA in WM165 cells after LEC interaction, needs to be recalculated. The findings look significant to the reviewers and therefore, the authors should carefully check whether the statistics have been carried out correctly. The Notch3 mRNA increase is in good agreement with the increase of Notch3 staining shown in Figure 2 and its quantification in panel F of Figure 2—figure supplement 2?

3) Figure 4D: the Results section states that Notch3 mRNA is decreased upon MMP14 depletion. There is a decrease that is, however, not significant. The text should be corrected accordingly.

Figure 4—figure supplement 2: panel B is unclear: were cells treated with siRNA against Notch3 to investigate the effect on MMP14 mRNA (is the increase significant or not? statistics missing?) or were cells treated with MMP14 siRNA to show the effect on Notch3, as said in the figure headline "MMP14 regulates Notch3"? Please clarify.

The actin blot in panel D displays unclear deformations. Is this a re-probing of the upper blot? It should be improved.

4) Figure 5: the increase of 12G10 staining in LEC-cocultured WM852 and WM167 cells (5A) as well as the decrease of sprouting after AIIB2 treatment (5B) are convincing. Therefore, it is sufficient to mention in the manuscript that FACS analysis for β1 integrin is not possible for the reasons expressed in the rebuttal letter.

The colocalization of the 12G10 and MMP14 signals is not convincing (5C). Please replace with a better image.

Although the effect of siRNA-mediated depletion of MMP14 on the decrease of 12G10 is convincing, the authors claim no change of total β1 integrin despite a decrease in β1 staining in 5F, which is not confirmed by the quantification shown in figure 5E. Is it possible that Figure 5F is not representative? Please clarify this discrepancy?

5) Figure 5—figure supplemental 2: the headline of the figure is inappropriate. Since this figure does not show experiments of NCID-mediated β1 integrin activation or sprouting, it would be better to change the headline to "β1 integrin activity does not affect MMP14 and Notch3 expression".

6) Figure 6D: the authors investigate WM793 cells (originally non-sprouting cells upon LEC co-culture). Treatment with NCID increases the invasive phenotype, whereas blocking β1 integrin curbs invasion indicating that β1 integrin is required for the invasive phenotype induced by active Notch3 (NCID). MMP14 inhibition does not block the invasive phenotype indicating that MMP14 is probably upstream.

Therefore, it would be more appropriate to change the title of Figure 6D to: “NCID overexpression provokes a β1 integrin-dependent 3D sprouting in WM793 cells".

Why did the authors use WM793 and not Bowe cells? Is this due to the Golgi-retained MMP14?

7) Author response image 3: the authors show that MMP14 or NCID is sufficient to induce invasive phenotype in WM cells. Why does neither MMP14 nor NCID overexpression in Bowe cells induce the invasive phenotype? Does it mean that it is not possible to induce an invasive phenotype after overexpression of MMP14 or NCID? The absent effect of MMP14 overexpression in Bowe cells is explained by the Golgi retention of MMP14 indicating that MMP14 cell surface localisation is needed (Figure 4—figure supplement 1). Therefore, overexpression of NCID should be sufficient if NCID is downstream to MMP14. What is missing? The reviewers suspect that β1 integrin needs to be rendered competent by MMP14 at the cell surface. The electrophoretic pattern of β1 integrin is similar in WM852 and WM793 cells before and after interaction with LEC, whereas the electrophoretic pattern of β1 integrin is different in Bowe cells as shown in Figure 5—figure supplement 1. The authors should discuss this conundrum in the revised manuscript.

---

## [Author Response]

Essential revisions:1) The link between MMP14, Notch3 and β1 integrin is not worked out well. Please test whether Notch3 depletion and/or inhibition alter MMP14 levels and β1 integrin activation.

i) We have now further addressed this important point by first silencing Notch3 in WM852* followed by measuring MMP14 levels by RTqPCR and IFA as well as β1 integrin activation by IFA. The results are now shown in Figure 4—figure supplement 2B-C for MMP14, and in Figure 5—figure supplement 2A-B for active β1-integrin, and demonstrated no significant change in MMP14 or active β1 integrin levels upon depletion of Notch3.

And whether inhibition of β1 integrins changes MMP14 and Notch3 levels and Notch target gene expression.

ii) This was addressed by treating WM852 and WM852* cells with the β1-integrin blocking antibody AIIB2 followed by measurement of the levels for MMP14, Notch3 and its downstream targets Hey1 and Hes1. RTqPCR revealed no changes in the expression levels of *MMP14, NOTCH3, HEY1 and HES1 after* AIIB treatment. This was further confirmed by IFA analysis for Notch3 and MMP14. These new results are now shown in Figure 5—figure supplement 2C, D, E.

2) MMP14 and β1 integrin levels should be quantified by FACS.

We agree with the reviewers that quantification of MMP14 and β1-integrin levels is more informative by FACS, and we have carried out FACS analyses using WM852 and WM852* cells using antibodies against MMP14, active β1-integrin and total β1-integrin.

The FACS analysis confirmed our finding obtained by IFA that upon the LEC co-culture WM852 show a significant increase in the cell surface MMP14 levels. This data is now included in Figure 4—figure supplement 1A.

However, despite three independent repetitions of the FACS analysis we were unfortunately not able to detect the increase in active β1-integrin levels in the WM852* (Author response image 1). As the activity of β1-integrin may be (depending on the activation mechanisms involved) linked to retained adhesion the LEC-induced increase in β1-integrin activation detected in the co-cultured cells by IFA could be sensitive to the cell detachment process (for which we used trypsinization, HyQtase or EDTA), whereas the MMP14 increase was retained during this process and allowed detection by FACS.

To further test this hypothesis we immunostained the magnetically separated WM852 and WM852* cells with both active β1-integrin antibodies 12G10, 9EG7 and MMP14 specific antibody and compared the staining patterns with cells grown in monotypic culture or LEC cocultured on coverlips, and obtained results similar to FACS (this experiment is shown in Author response image 1), i.e. no increase in active β1-integrin was observed in the WM852* after detachment from co-culture, magnetic separation and replating on glass coverslips. However, if the cells were embedded into 3D fibrin right after separation, the increase in active β1-integrin signal form 12G10 antibody was retained (Author response image 1) suggesting that the 3D fibrin enabled to sustain the activation state of the integrin. This also explains the sensitivity of the LEC-induced 3D sprouting phenotype to AIIB2 treatment.

**Author response image 1. respfig1:** (**a**) Flow cytometry analysis of active b1-integrin levels in WM852 and WM852* using 12G10 (upper) and 9EG7 (lower) antibodies. (**b**) Confocal images of WM852* replated after separation on coverslips for 24h (left panels), WM852 +LEC co-culture (not subjected to magnetic separation; middle panels) and monotypic WM852 (right panels). Cells were stained with the indicated antibodies (red), nuclei were counterstained with Hoechst 33342. Scale bar: 20mM. (**c**) Representative confocal z-stack projections of 3D fibrin assay after magnetic separation of WM852 cells co-cultured with LECs (*,right panel) or from monotypic culture (left panel). The GFP expressing (green) melanoma cells were stained with 12G10 antibody (red), nuclei were counterstained with Hoechst 33342 (blue). Scale bar = 50 µm.

Immunostaining is not quantitative.

To respond to this concern raised by the reviewers I have explained below the quantification process in detail.

To ensure that the image analysis was accurate and precise we took the following steps:

1) In each experiment, more than 100 cells were analysed per condition.

2) We acquired our images as 16-bit. This format provides the largest possible scale of intensities and gives the possibility to detect even small differences in fluorescence intensity.

3) Laser powers were adjusted to make sure no pixel saturation occurred during imaging and all the stainings to be compared were taken with the same exposures and settings of the microscope.

4) Each confocal stack contained the entire thickness of all the imaged cells so that no portion of any cells was left outside the imaged field.

5) In each confocal stack, the optical thickness between one image and the next was reduced to <300 nm, far below the resolution limit of the microscope along the vertical (Z) axis. This way no empty (i.e. non-imaged) gaps were left between each image.

6) Images were not adjusted prior to quantification.

7) After Z-projection of the max-intensities, the intensity of fluorescence in each set of images was quantified using the image analysis software (Cellprofiler; Carpenter et al., Genome Biol. 2006 &). This method is completely unbiased, widely used to obtain reproducible quantitative data from large data sets, such as image-based screens, to single images.

And furthermore, shows an unclear perinuclear accumulation of MMP14. What does the perinuclear location mean?

The cells in Figure 4 and c indeed show perinuclear localization of MMP14, especially prominent in the Bowes cell line. To study the location of the perinuclear MMP14, we have costained Bowes and WM852 cells with antibodies targeting MMP14 and TGN46 which defines the location of trans-Golgi network. As now shown in Figure 4—figure supplement 1C, the perinuclear signal of MMP14 co-localized with TGN46, especially in Bowes.

The corresponding Western blot for Figure 5—figure supplement 1C should be shown.

As requested, we performed Western blot analysis for total β1-integrin of mono/co-cultured melanoma cell lines. Since the P5D2 antibody, used for the IFA staining in Figure 5C, does not work in Western blotting, we had to use the Abcam mouse monoclonal antibody 52971. The Western blot is included in Figure 5—figure supplement 1E (and an independent repetition shown as Author response image 2), and reveals that LEC-co-culture induces in WM852*, WM165* and WM793* a decrease in the upper β1-integrin specific band (mature form of the integrin), and an increase in a faster migrating band, probably representing newly synthesized, immature β1-integrin. This was particularly pronounced in WM852* compared to WM852.

However, neither a decrease nor a change in β1-integrin localization were observed in the IFA analyses (Figure 5—figure supplement 1C-D).Therefore, to understand this discrepancy and assess the total β1-integrin levels, we analysed the levels of total β1-integrin in WM852 and WM852* by flow cytometry using two different antibodies; P5D2 (already used in IFA) and K20 antibodies (since the Abcam 52971 used for Western blot it is not suitable for FACS). As shown in the Author response image 2 we could not observe any change in the total β1-integrin levels with the P5D2 antibody (thus confirming the IFA data obtained with this same antibody) whereas with the K20 antibody we could detect about 29% decrease. Given these results, we cannot exclude that upon co-culture, especially in WM852*, the total β1-integrin level and/or maturation and trafficking processes might be altered. However, further studies would be needed to clarify this issue.

**Author response image 2. respfig2:** (**a**) Left panel: immunoblot of indicated cells lines cultured with or without LECs and separated by magnetic separation and stained with indicated antibodies. Right panel: quantification of relative β1-integrin band intensities from two independent experiments as shown in left panel. The signal intensity was normalized to actin. The average is shown, error bars represent SD. (**b**) Flow cytometry analysis of total β1-integrin levels in WM852 and WM852* using P5D2 (left) and K20 antibodies. Mean fluorescence intensity normalized to WM852 is shown, error bars represent SD.

And β1 integrin activation verified with the rat mAb 9EG7.

As suggested by the reviewers, the activation of β1-integrin was verified using the 9EG7 antibody by immunostaining. In agreement with the 12G10 antibody staining, we observed a significant increase in the 9EG7 signal intensity in WM852* cells. These results have been included in Figure 5—figure supplement 1B. In addition, we also decided to use the 9EG7 antibody to assess whether silencing Notch3 would affect β1-integrin activation (as requested in reviewer point 1.i) and presented in Figure 5—figure supplement 2B).

3) The dataset derived from a comparison of melanoma cell lines with and without LEC priming is interesting and the list(s) of genes with altered expression should be shown as supplementary data. Furthermore, analyze and describe in a few sentences (or an additional figure panel) the genes and/or pathways altered in both cell lines after LEC priming and the genes specific for WM852 or Bowes cells.

As requested, we now provide the lists of genes with altered expression upon co-culture for WM852 and Bowes as Supplementary file 1- related to Figure 2 (more than 2-fold significantly altered genes) and Supplementary file 2 – related to Figure 2 (all genes significantly up- and downregulated listed for both cell lines in the selected pathways). We also show the pathways altered in both cell lines after LEC priming and the pathways and genes specific for WM852 or Bowes cells (New panels in Figure 2A-C). Description of the added data is provided in the Results text (subsection “Interaction with LECs induces transcriptional changes in melanoma gene expression”).

4) To confirm validity of the pathway induced by LECs, test whether the sprouting phenotype is induced in Bowes and WM793 cells by upregulating MMP14 on the cell surface or by activating Notch3 or β1 integrin signaling.

To respond to this request, we transiently transfected both WM793 and Bowes with expression plasmids encoding MMP14 and the constitutively active domain of Notch3, NICD3, and activated β1-integrin in WM793 using 12G10 coating on the culture dish. These treatments were followed by the 3D sprouting assay in fibrin for 96 h.

Unexpectedly, ectopic expression of MMP14 was toxic and induced cell death in WM793 already in 2D and we could not proceed for the 3D sprouting assay. However, cell viability was not affected by NICD3 expression using the same transfection conditions (Author response image 3 and Figure 6B), and the subsequent 3D fibrin assay revealed an increase in the sprouting phenotype upon NICD3 ectopic expression (new Figure 6B). In contrast, no change in the 3D growth phenotype was observed upon b1-integrin activation by 12G10 treatment (new Figure 6B-C). To further investigate the role of MMP14 and β1-integrin in the NICD3induced sprouting we treated the NICD3-expressing WM793 with the MMP14 specific inhibitor NSC405020 and a b1-integrin blocking antibody AIIB2. While the former had no effect on the sprouting phenotype, thus confirming MMP14 to be upstream of Notch3 activation, the b1integrin blocking antibody inhibited the sprouting (new Figure 6D). These new data are presented in lines (subsection “NICD3 ectopic expression is sufficient to induce 3D sprouting in non-metastatic WM793 cells”).

In Bowes cells, ectopic expression of MMP14 and NICD3 did not affect the viability of the cells (Author response image 3). However, the singly invading Bowes cells did not change their mode of invasion upon expression of MMP14 or NICD3 (Author response image 3). We refer to this negative data as ‘data not shown’.

**Author response image 3. respfig3:** (**a**) Confocal images of WM793 transiently transfected with MMP14 expressing plasmid (left) or a control vector (right) stained with Notch3 (green) and MMP14 (red) antibodies, nuclei were counterstained with Hoechst 33342. (**b**) Confocal imaged of Bowes transiently transfected with MMP14 (left), NICD3 (middle) expressing plasmid, or a control vector (right) stained with Notch3 (green) and MMP14 (red) antibodies, nuclei were counterstained with Hoechst 33342. (**c**) Representative confocal images of 3D fibrin assays of Bowes cells treated as in (**b**). GFP expressing (green) melanoma cells were stained with Phalloidin A594 (red), nuclei are counterstained with Hoechst 33342 (blue). Maximum intensity Z-projections of the confocal stacks are shown. Scale bars= 50 µm.

5) Verify the involvement of Notch3 and MMP14 in the in vivo metastatic dissemination of 3D LEC-co-cultured melanoma cells.

To address this important point, we decided to use the zebrafish xenograft model as human melanoma cells have been shown to retain their invasive in vivo behaviour when transplanted to zebrafish embryos (Chapman, Fernandez del Ama et al., 2014). The zebrafish xenograft experiments and analyses were conducted in collaboration with and supervision by Dr. Ilkka Paatero, the head of zebrafish core at the Turku Centre for Biotechnology (University of Turku, Finland), and he is therefore included as a new co-author in the manuscript. The results of the experiment are now included in a new Figure 7 and show that depletion of Notch3 or MMP14 reduced the enhanced metastatic properties of WM852*.

6) The LEC/tumor cell interaction should be better characterized with the available transcriptomic data to potentially identify a cadherin switch, adhesion receptor switch, homo or heterotypic interaction, needs of specific ECM, etc.The text says that "global gene expression revealed change in ECM-receptor interaction". What is meant with this sentence?

We have further analysed our transcriptomic data which revealed intriguing differences in the genes enriched in WM852 and Bowes following LEC co-culture. We did not observe a classical cadherin or adhesion receptor switch, but did uncover significant differences between the WM852* and Bowes* cells in the genes functioning in the ECM-receptor interaction. Moreover, we also found genes enriched only in Bowes cells that are involved in cell-cell and cell-matrix contacts, like focal adhesion, TGF-b signalling and tight junction pathways. These changes are now highlighted and shown in Figure 2A-C and as new panels in Figure 2—figure supplement 2A-B). We have also clarified the text saying that "global gene expression revealed change in ECM-receptor interaction". The new text can be found in the subsection “Interaction with LECs induces transcriptional changes in melanoma gene expression”.

[Editors' note: further revisions were requested prior to acceptance, as described below.]

The manuscript has been improved but there are some remaining issues that need to be addressed, as outlined below:It is shown that the LEC-induced metastatic melanoma 3D growth phenotype is Notch3- and β1 integrin-dependent. Despite the evidence showing that MMP14 is upstream of Notch3 and β1 integrin, the authors do not provide convincing experimental data for a MMP14-Notch3- β1 integrin axis, as claimed in the second paragraph of the Discussion. It is demonstrated that β1 integrin is needed for dissemination once Notch3 is activated. However, evidence for a direct effect of Notch3 on β1 integrin activity is missing. Instead, the findings shown in the manuscript suggest that MMP14 acts separately on Notch3 and β1 integrin. This issue requires clarification.

We agree that the some of our text in the original, revised manuscript file was misleading, and we have therefore rephrased and tried to explain better what the results are showing (see subsection “NICD3 ectopic expression is sufficient to induce 3D sprouting in non-metastatic WM793 cells”).

In addition, the statistics are incomplete and the quality of some images requires improvement.

See below.

1) Statistics have to be provided for all graphs. This is essential.

We have now added the required statistics for all graphs as requested. To strengthen the power and significance of the statistical analysis, we have repeated a few of the experiments such as the time course shown in Figure 3C.

2) Figure 2—figure supplement 2: the lack of significance in panel G, which shows an increase of Notch3 mRNA in WM165 cells after LEC interaction, needs to be recalculated. The findings look significant to the reviewers and therefore, the authors should carefully check whether the statistics have been carried out correctly. The Notch3 mRNA increase is in good agreement with the increase of Notch3 staining shown in Figure 2 and its quantification in panel F of Figure 2—figure supplement 2?

The statistics for Figure 2—figure supplement 2G has now been checked carefully. The quantification represents an average of four independent experiments. Although each experiment showed an increase in the *NOTCH3* mRNA levels in the LEC primed WM165* compared to the WM165 from monotypic cultures, the level of increase has varied markedly between the experiments, and therefore significance was not achieved (p=0.1).

3) Figure 4D: the Results section states that Notch3 mRNA is decreased upon MMP14 depletion. There is a decrease that is, however, not significant. The text should be corrected accordingly.

We have now corrected the sentence accordingly (subsection “MMP14 is required for the invasively sprouting 3D growth of LEC primed melanoma cells”, last paragraph).

Figure 4—figure supplement 2: panel B is unclear: were cells treated with siRNA against Notch3 to investigate the effect on MMP14 mRNA (is the increase significant or not? statistics missing?) or were cells treated with MMP14 siRNA to show the effect on Notch3, as said in the figure headline "MMP14 regulates Notch3"? Please clarify.

The cells were treated with siRNAs targeting either *NOTCH3 or HEY1* or with control siRNA (siCtrl) or left untreated. The relative mRNA levels of *MMP14* were compared to the untreated control cells. These results indicate that *NOTCH3* depletion has a small but non-significant effect on *MMP14* mRNA and *HEY1* silencing has no effect. The statistics are now added to the figure panel, and the figure headline was changed to a more appropriate one: “Notch3 does not regulate MMP14 expression, but MMP14 positively regulates Notch3”.

The actin blot in panel D displays unclear deformations. Is this a re-probing of the upper blot? It should be improved.

We are now providing a new western blot for the panel D to improve the band quality for actin and Notch3/NICD3. The quantification of this immunoblot has been added to the right panel as well.

4) Figure 5: the increase of 12G10 staining in LEC-cocultured WM852 and WM167 cells (5A) as well as the decrease of sprouting after AIIB2 treatment (5B) are convincing. Therefore, it is sufficient to mention in the manuscript that FACS analysis for β1 integrin is not possible for the reasons expressed in the rebuttal letter.

As proposed by the reviewers, the reasons for the unsuccessful FACs analyses for active β1-integrin are now added in the first paragraph of the subsection “Change in the 3D growth of LEC primed melanoma cells is integrin dependent”.

The colocalization of the 12G10 and MMP14 signals is not convincing (5C). Please replace with a better image.

A new confocal image with high resolution is provided for Figure 5C.

Although the effect of siRNA-mediated depletion of MMP14 on the decrease of 12G10 is convincing, the authors claim no change of total β1 integrin despite a decrease in β1 staining in 5F, which is not confirmed by the quantification shown in figure 5E. Is it possible that Figure 5F is not representative? Please clarify this discrepancy?

We agree that the image in Figure 5F may not have been the most representative; therefore we have now replaced the panels with ones that better represent the mean values of the intensity measurements for the quantification.

5) Figure 5—figure supplemental 2: the headline of the figure is inappropriate. Since this figure does not show experiments of NCID-mediated β1 integrin activation or sprouting, it would be better to change the headline to "β1 integrin activity does not affect MMP14 and Notch3 expression".

We thank the reviewers for pointing out this unfortunate mistake and have corrected the headline as suggested.

6) Figure 6D: the authors investigate WM793 cells (originally non-sprouting cells upon LEC co-culture). Treatment with NCID increases the invasive phenotype, whereas blocking β1 integrin curbs invasion indicating that β1 integrin is required for the invasive phenotype induced by active Notch3 (NCID). MMP14 inhibition does not block the invasive phenotype indicating that MMP14 is probably upstream.Therefore, it would be more appropriate to change the title of Figure 6D to: “NCID overexpression provokes a β1 integrin-dependent 3D sprouting in WM793 cells".Why did the authors use WM793 and not Bowe cells? Is this due to the Golgi-retained MMP14?

We thank the reviewers for suggesting a more appropriate title for Figure 6, which is now changed to the figure legends. The experiments were actually done with both WM793 and Bowes cells; however, exogenous expression of NICD3 or MMP14 did not induce a change in the 3D growth phenotype of Bowes cells. The lack of phenotype in Bowes cells was described in the original, revised manuscript in the Results section (subsection “NICD3 ectopic expression is sufficient to induce 3D sprouting in non-metastatic WM793 cells”, first paragraph).

7) Author response image 3: the authors show that MMP14 or NCID is sufficient to induce invasive phenotype in WM cells. Why does neither MMP14 nor NCID overexpression in Bowe cells induce the invasive phenotype? Does it mean that it is not possible to induce an invasive phenotype after overexpression of MMP14 or NCID? The absent effect of MMP14 overexpression in Bowe cells is explained by the Golgi retention of MMP14 indicating that MMP14 cell surface localisation is needed (Figure 4—figure supplement 1). Therefore, overexpression of NCID should be sufficient if NCID is downstream to MMP14. What is missing? The reviewers suspect that β1 integrin needs to be rendered competent by MMP14 at the cell surface.

As mentioned in the Discussion of the previous, revised version of the manuscript the Bowes cells already have an invasive, single cell 3D sprouting phenotype, but do not grow as expansive cell colonies as WM793, WM852 and WM165. Therefore, it is possible that Bowes cannot be induced to this collective-type invasive phenotype, as it may require the expansive growth mode as a prerequisite for the LEC-mediated Notch3 dependent change in the 3D growth. However, we do not think that the lack of phenotype change in Bowes would be due to MMP14 not trafficking onto the plasma membrane, as we can see MMP14 plasma membrane localization upon its ectopic expression (see Author response image 3 from our previous Response to Reviewers).

The electrophoretic pattern of β1 integrin is similar in WM852 and WM793 cells before and after interaction with LEC, whereas the electrophoretic pattern of β1 integrin is different in Bowe cells as shown in Figure 5—figure supplement 1. The authors should discuss this conundrum in the revised manuscript.

The resemblance of the electrophoretic pattern of integrin β1 in WM852 and WM793 if compared to the Bowes cells may indeed indicate differences in integrin β1 processing and trafficking in Bowes and WM852/793. Thus, this could partially explain why ectopic expression of NICD3 was sufficient to induce invasive phenotype in WM793 but not in Bowes. As suggested by the reviewer, we have now discussed this conundrum in the Results section (subsection “Change in the 3D growth of LEC primed melanoma cells is integrin dependent”, first paragraph) and also in Discussion (second paragraph).

Note: One of the co-authors, Krista Ojala, just recently (as of March 12) changed her last name due to personal reasons from Ojala to Tuohinto. Her new last name is now indicated in the author list in red.